# Systemic viral spreading and defective host responses are associated with fatal Lassa fever in macaques

Nicolas Baillet [1,2], Stéphanie Reynard [1,2], Emeline Perthame [3], Jimmy Hortion[1,2], Alexandra Journeaux[1,2], Mathieu Mateo [1,2], Xavier Carnec[1,2], Justine Schaeffer[1,2], Caroline Picard[1,2], Laura Barrot [4], Stéphane Barron[4], Audrey Vallve[4], Aurélie Duthey[4], Frédéric Jacquot[4], Cathy Boehringer[4], Grégory Jouvion [5], Natalia Pietrosemoli[3], Rachel Legendre [3], Marie-Agnès Dillies[3], Richard Allan[6], Catherine Legras-Lachuer[6], Caroline Carbonnelle[4], Hervé Raoul [4] & Sylvain Baize [1,2✉]

Lassa virus (LASV) is endemic in West Africa and induces a viral hemorrhagic fever (VHF) with up to 30% lethality among clinical cases. The mechanisms involved in control of Lassa fever or, in contrast, the ensuing catastrophic illness and death are poorly understood. We used the cynomolgus monkey model to reproduce the human disease with asymptomatic to mild or fatal disease. After initial replication at the inoculation site, LASV reached the secondary lymphoid organs. LASV did not spread further in nonfatal disease and was rapidly controlled by balanced innate and T-cell responses. Systemic viral dissemination occurred during severe disease. Massive replication, a cytokine/chemokine storm, defective T-cell responses, and multiorgan failure were observed. Clinical, biological, immunological, and transcriptomic parameters resembled those observed during septic-shock syndrome, suggesting that similar pathogenesis is induced during Lassa fever. The outcome appears to be determined early, as differentially expressed genes in PBMCs were associated with fatal and non-fatal Lassa fever outcome very early after infection. These results provide a full characterization and important insights into Lassa fever pathogenesis and could help to develop early diagnostic tools.

---

[1] Unité de Biologie des Infections Virales Emergentes, Institut Pasteur, Lyon, France. [2] Centre International de Recherche en Infectiologie (CIRI), Université de Lyon, INSERM U1111, Ecole Normale Supérieure de Lyon, Université Lyon 1, CNRS UMR5308, Lyon, France. [3] Hub de Bioinformatique et Biostatistique – Département Biologie Computationnelle, Institut Pasteur, USR 3756 CNRS, Paris, France. [4] Laboratoire P4 INSERM – Jean Mérieux, INSERM US003, Lyon, France. [5] Neuropathologie Expérimentale, Département de Santé Globale, Institut Pasteur, Paris, France. [6] ViroScan3D SAS, Trévoux, France. ✉email: Sylvain.baize@pasteur.fr

Lassa Fever is a viral hemorrhagic fever that is endemic in Nigeria and the Mano River Union countries (Sierra Leone, Guinea, and Liberia), with sporadic cases in bordering West African countries and recent outbreaks in Benin[1,2]. Lassa fever represents a significant cause of morbidity and mortality, with tens of thousands of cases annually and thousands of fatalities. The main natural host of Lassa virus (LASV), a Mammarenavirus of the Arenaviridae family, is the widespread and commensal rodent *Mastomys natalensis*, but alternate rodent reservoirs have been described[3]. Rodent-to-human transmission of LASV generally occurs after the inhalation of contaminated dust or cutaneous-mucosal contact with material soiled by rats, and human-to-human transmission is then observed[4]. Approximately 20–40% of hospitalized patients succumb to infection. No licensed vaccine or treatment with demonstrated efficacy is currently available. Lassa fever has been recently listed by the WHO as an epidemic threat requiring urgent preparedness.

The diverse LASV strains are grouped into seven major lineages based on their geographical location. Lineages I, II, and III circulate in Nigeria and lineage IV strains were isolated in the Mano River Union countries of Guinea, Sierra Leone, and Liberia. Three additional lineages have recently emerged in Mali, Cote d'Ivoire, Ghana, Togo, and Benin[2,5–7]. The two strains used in this study, AV and Josiah, belong to lineage V and IV, respectively, and are known to induce a severe disease in humans and non-human primates (NHPs)[8,9]. The AV strain has been isolated in a fatally-infected patient in 2000, whereas the Josiah strain represents the prototypic and reference LASV strain used in almost all studies. These strains are phylogenetically close as they share ~82% of nucleotide homology and 95% of amino acids identity, based on the S segment[9]. A large number of Lassa fever patients experience asymptomatic infections or mild symptoms, such as fever, headache, and asthenia. In other patients, fever, asthenia, anorexia, abdominal pain, nausea, vomiting, diarrhea, cough, and sore throat are observed. Edema of the face and neck, conjunctivitis, hemorrhage, and acute respiratory distress then occur in severe patients. The terminal stage is characterized by multiorgan failure and terminal shock[10]. In survivors, sequelae may appear after recovery, including sensorineural hearing loss[11]. The location of endemic areas and the difficulty in accessing patients have stifled the investigation of Lassa fever in humans. Our current understanding of the pathogenesis and immune response comes from animal models but is still limited. Animals, such as immunocompromised mice and strain 13 guinea pigs, are widely used to screen vaccines and therapeutics, but they fail to mirror human disease[12–15]. Only NHPs fully mimic the clinical signs, viral tropism, immune responses, pathophysiological changes, and various outcomes observed in humans[8,16–21]. LASV infection is believed to begin with the targeting of macrophages and dendritic cells (DCs), with subsequent spreading to other cells[22]. However, direct viral cytopathic effects are not sufficiently severe to account for the observed vascular permeability and multiorgan failure. Thus, the cascade of pathogenic events that leads to terminal shock and death is still poorly known. Similarly, the immune mechanisms that allow patients to clear LASV infection and recover from Lassa fever have not been identified. Although macrophages and DCs may be the first target cells, they are not activated and do not secrete high amounts of cytokines. Low titers of neutralizing antibodies have been observed in NHPs and patients only after recovery and therefore do not correlate with survival. However, we and others have demonstrated that early activation and strong proliferation of T cells are associated with the control of LASV and recovery from Lassa fever in cynomolgus monkeys[8,23,24]. A robust T-cell response was also correlated with viremia resolution in a human case of Lassa Fever[25]. Here, we further explored the pathogenesis of Lassa fever and the immune responses induced by LASV by infecting cynomolgus monkeys with two different LASV strains: Josiah and AV, using the same dose and route of inoculation. Subcutaneous infection with LASV-AV resulted in nonfatal Lassa fever with moderate clinical signs, whereas all the animals infected with LASV-Josiah succumbed to Lassa fever. Thus, we performed a full characterization of Lassa fever pathophysiogenesis in an NHP model by analyzing clinical pathology, viral tropism and dynamics, histopathological changes, cytokine profile, immune responses, and transcriptomic signatures that are associated with fatal and nonfatal Lassa fever during the course of infection and disease.

## Results

**Clinical parameters and viral spreading during Lassa fever.** Cynomolgus monkeys were subcutaneously inoculated at the back of the thigh with 1000 focus-forming units (FFU) of LASV-AV ($n = 4$) or LASV-Josiah ($n = 6$) (Fig. S1a), whereas three were mock-infected. The animals were then monitored for 28 days and attributed clinical scores based on body temperature, weight, food and water intake, behavior, and clinical signs, with an endpoint score of 15. All Josiah-infected animals presented clinical scores that increased relentlessly from day 3 and were killed 11–15 days post infection (DPI) (Fig. 1a). They presented severe symptoms, with a high fever from day 4 and hypothermia at the terminal stages, asthenia, piloerection, anorexia, and continuous weight loss (Fig. S1b). One animal presented epistaxis on the day of euthanasia, whereas abdominal extravasation was detected in two others at necropsy. AV-inoculated animals experienced only mild clinical signs with no fever and a moderate clinical score due to weight loss and recovered by ~15 DPI (Fig. 1a). C-reactive protein (CRP) levels in the plasma of LASV-infected animals increased (Fig. S1c). Viral RNA appeared in the plasma of infected animals as soon as three DPI and infectious particles were first detected in Josiah- and AV-infected animals at 6 and 8 DPI, respectively (Fig. 1b). Viremia peaked at 12 DPI and then decreased in survivors, whereas it increased relentlessly in the Josiah-infected animals. Similar results were obtained in bone marrow (BM) samples, except that viral RNA was still detectable in the survivor samples 28 DPI (Fig. S1 d, e).

Additional groups of three animals were also infected with LASV-AV or LASV-Josiah and sequentially killed 2, 5, and 11 (only AV for this last timepoint) DPI to obtain access to the organs during the incubation period, at the onset of viremia, and at the peak of the disease (Fig. S1a). We also quantified the viral loads in organs obtained from the sequentially killed animals. Viral RNA was detected in the skin, close to the inoculation site, of all Josiah- and AV-infected animals as soon as 2 DPI, as well as in the spleen, inguinal lymph nodes (ILNs), and thymus of one and two, respectively, Josiah-infected animals and in the mesenteric lymph nodes (MLNs) of two of three AV-infected NHPs (Fig. 1c). Infectious particles were also detected in the skin of almost all infected animals. Five days after infection, the highest amounts of viral RNA were found for all animals in the skin, ILNs, MLNs, and spleen. The liver, cerebellum, kidneys, adrenal glands, intestines, lungs, thymus, and pancreas also contained viral RNA. At this timepoint, we detected infectious particles in the secondary lymphoid organs (SLOs), skin, and liver of all Josiah- and most AV-infected animals, whereas we detected LASV in the kidneys, adrenal glands, large intestine, lungs, reproductive system, and pancreas of Josiah-infected animals only. At 11 DPI, corresponding to the peak of the disease, all assessed tissues contained viral RNA. However, viral loads were 10- to 1000-fold higher in organs after infection with LASV-Josiah than with LASV-AV. In AV-infected

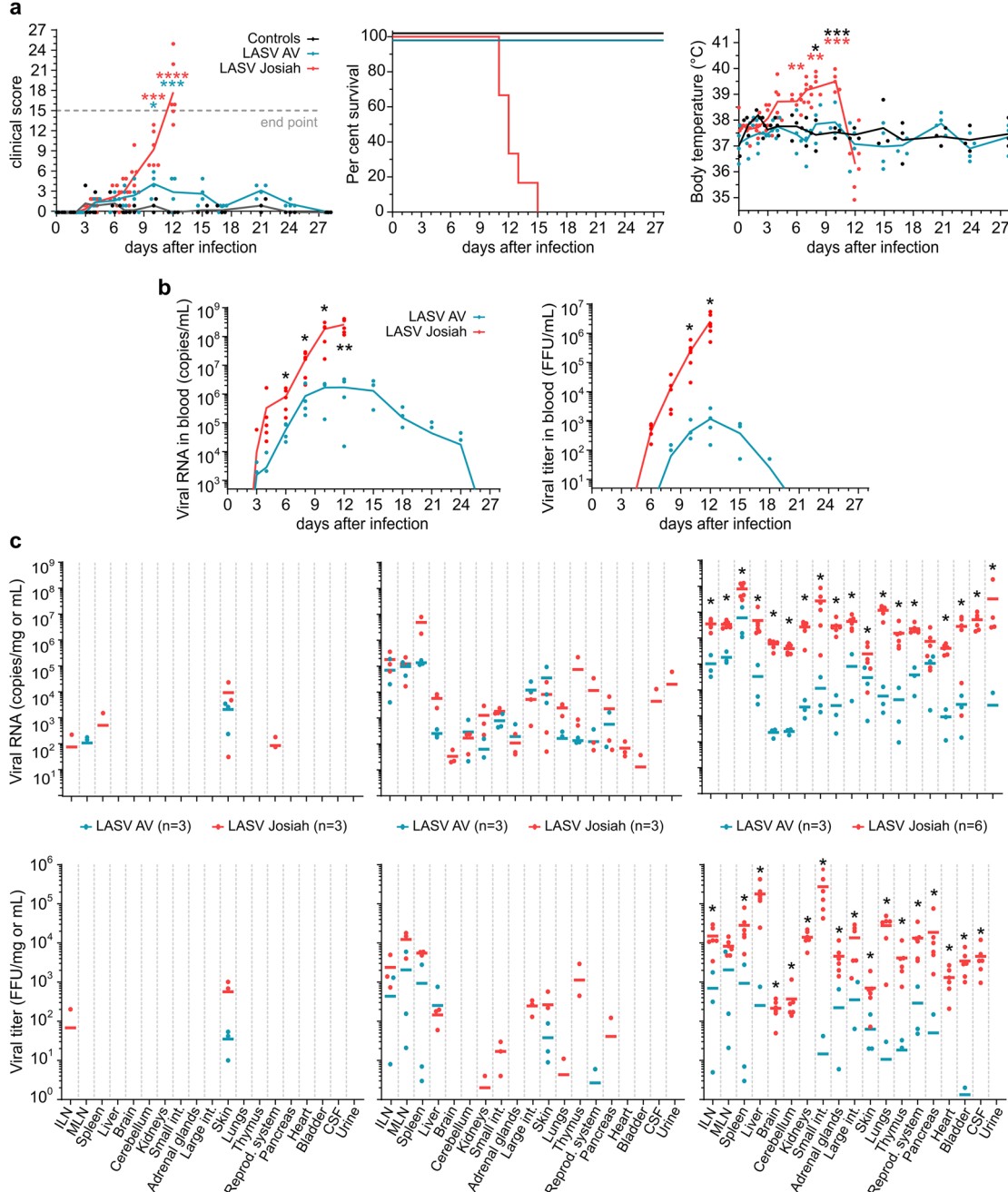

**Fig. 1 Clinical and virological features of Lassa fever. a** Clinical score, survival rate, and monitoring of body temperature are presented. Results show mean values and individual data for each cohort, except for survival rate, represented with a Kaplan–Meier curve. Controls ($n = 3$), LASV-AV ($n = 4$), LASV-Josiah ($n = 6$). *$p < 0.05$, **$p < 0.01$, ***$p < 0.001$, ****$p < 0.0001$, as determined by a one-way ANOVA multiple comparisons test for analysis between controls and AV- and Josiah-infected animals and by a $t$ test for comparison between only controls and AV-infected NHPs. Red asterisks indicate a significant difference between LASV-Josiah and controls, blue asterisks a significant difference between LASV-AV and controls, and black asterisks a significant difference between LASV-Josiah and LASV-AV. **b** Quantification of viral load and infectious particles in the plasma of animals according to the time after LASV infection. Mean values of each cohort and individual data points are presented as the number of viral RNA copies/ml (viral load) or FFU/ml (infectious particles) according to the time after challenge with AV (blue dots) or Josiah (red dots). LASV-AV ($n = 4$), LASV-Josiah ($n = 6$). The individual data at the time before the death of all LASV-Josiah-infected primates were used to calculate the mean determined at 12 DPI *$p < 0.05$, **$p < 0.01$, as determined by a Student's $t$ test. **c** Quantification of viral RNA copies and infectious particles per milligram of various tissues, including inguinal lymph nodes (ILN), mesenteric lymph nodes (MLN), small intestine (Small int.), large intestine (Large int.), the reproductive system (ovaries or testicles) (Reprod. system), and cerebrospinal fluid (CSF) at 2 (left panel), 5 (middle panel), and 11 (right panel) DPI. Individual and mean values are presented for the two cohorts. *$p < 0.05$, as determined by a Student's $t$ test.

animals, the presence of infectious particles in the lymphoid organs, liver, adrenal glands, intestines, skin, lungs, reproductive system, and pancreas was heterogeneous. In contrast, all organs of Josiah-infected NHPs contained very high LASV titers, particularly the

adrenals and central nervous system (CNS). In these animals, the viral RNA to titer ratio, an indicator of the efficiency of viral replication, ranged between 100 and 1000 for most tissues, except for brain and cerebellum, for which it was ~3000, and liver and

pancreas, for which the ratio was ~30 (Fig. S1f). Viral RNA was still present 28 DPI in the tissues of AV-infected animals (Fig. S1g).

**Histopathological features of Lassa fever**. The earliest pathological events were mostly detectable at 5 DPI in the SLOs and lungs of infected NHPs. In the LNs, AV-, and Josiah-infected animals exhibited an alteration of the cortex architecture, characterized by disorganization or hyperplasia of the lymphoid follicles (Fig. S2a middle and right panel). At 11 DPI, the loss of follicle architecture was still observable in AV- and Josiah-infected animals, often with hyperplasia of macrophages, sometimes containing phagocytized apoptotic bodies (tingible-body macrophages; see insets). LASV was frequently observed 5 DPI in non-lymphocytic cells in all infected animals and was still observable in LNs 11 DPI (Fig. 2a). AV-infected cynomolgus monkeys also exhibited apoptotic lymphocytes in the thymus as soon as 5 DPI (Fig. S2b middle panel). There was also massive lymphocyte apoptosis 11 DPI in the thymus of Josiah-infected animals (Fig. S2b right panel). LASV was detected at 5 and 11 DPI in the thymus of all infected NHPs (Fig. 2b). In the spleen, amyloid protein was present in the center of lymphoid follicles of all Josiah-infected animals and in one AV-infected animal (Fig. S2d inset). There were numerous tingible-body macrophages in the Josiah-infected NHPs. Accordingly, LASV was observable in the spleen as soon as 5 DPI, and more frequently at 11 DPI, in both AV- and Josiah-infected animals (Fig. 2c). However, the distribution of LASV antigen was much more widespread in the Josiah-infected animals. We did not observe histopathological changes in the lungs of the AV-infected NHPs, either at 5 or 11 DPI, with the exception of one of three animals at each timepoint that presented minor foci of interstitial pneumonitis (Fig. S2c middle panel). This was more pronounced in the Josiah-infected animals, with infiltration of mononucleated cells, resulting in thickening of the alveolar walls prior to death at 11 DPI (Fig. S2c right panel). LASV was also present in the alveolar cells of Josiah-infected animals 5 and particularly 11 DPI (Fig. 2d). Although we detected minor evidences of inflammation in the liver of AV-infected NHPs, the Josiah-infected animals showed portal to more extensive hepatitis (with apoptosis of hepatocytes for one out of three animals) (Fig. S2e). These observations are in accordance with the strong widespread LASV immunopositive staining throughout the hepatic parenchyma and observed only in Josiah-infected NHPs (Fig. 2e). We observed similar inflammation patterns in the kidneys in the vicinity of arteries from the cortico-medullary junction in Josiah- but not AV-infected animals (Fig. S2f). Accordingly, LASV was detectable in cells from the cortico-medullary junction in Josiah- but not AV-infected NHPs (Fig. 2f). We detected foci of LASV-positive adrenal cortical cells in the zona glomerulosa and zona fasciculata of all Josiah- but not AV-infected animals (Fig. 2g). Although probably nonspecific, the cerebral cortex of all infected animals showed edema. However, at 11 DPI, we detected perivascular cuffing and microglial activation, i.e., signs of cerebral inflammation encountered in severe forms of the disease, in the brain of Josiah-infected CMs (Fig. S2g). Histopathological features were also present during recovery 28 DPI in some AV-infected animals. We indeed detected inflammatory infiltrates centered on (i) hepatic portal tracts (Fig. S2h), (ii) renal cortico-medullary junction and medulla (Fig. S2i), (iii) myocardial arteries (Fig. S2j) and (iv) 2 cerebral blood vessels (perivascular cuffing, Fig. S2k).

**Biochemical analytes and soluble mediators during Lassa fever**. There was a slight but significant increase in the concentration of aspartate aminotransferase (AST) and alanine aminotransferase (ALT) in survivors at 10 DPI, which returned to basal values 21 DPI. AST and ALT levels increased earlier and were much higher until death in Josiah-infected cynomolgus monkeys (Fig. 3a). High concentrations of lactate dehydrogenase (LDH) were present in the plasma of the animals that died from day 6, increasing steadily until death. In contrast, the increase in LDH levels occurred later and was moderate in survivors. The level of albumin (ALB) dropped in both groups, with the minimum values at the peak of the disease in the survivors. Creatinine and urea levels increased substantially in Josiah-infected NHPs. We observed low levels of $Na^+$ and $Cl^-$ in the plasma at the terminal stages of the animals that died.

We then have set out the profiles of circulating soluble mediators in infected animals. Concerning pro-inflammatory cytokines, plasma levels of IL-6, TNFα, and IL12/23 started rising from 6 to 8 DPI and continued rising until death at 12 DPI in Josiah- but not AV-infected animals (Fig. 3b). There was a transient release of IFNα, peaking at 6 DPI, for all infected animals. IL-15 levels increased in animals starting from 2 DPI and continued to rise until death for the Josiah-infected NHPs, whereas they returned to basal values 15 DPI for the AV-infected animals. The levels of anti-inflammatory cytokines IL-1RA and IL-10 increased dramatically from 6 DPI to death in the Josiah-infected animals, whereas there was only a moderate increase in AV-infected NHPs from 6 to 15 or 28 DPI for IL-1RA and IL-10, respectively (Fig. 3c). Monocyte chemoattractant protein (MCP) 1 and macrophage inflammatory protein (MIP) 1β concentrations slightly and transiently increased in AV-infected animals, whereas high and increasing levels were measured in the terminal stages for the Josiah-infected animals (Fig. 3d). IL-8 was transiently secreted at 1 DPI in the Josiah-infected NHPs, whereas its release was delayed and more sustained in the AV-infected animals. We finally assessed different soluble mediators involved in cytotoxic activities and regulation of the immune response. We detected elevated levels of soluble (s) CD137 and perforin, peaking at 12 DPI in all animals before returning to basal values at ~18 DPI in the AV-infected animals (Fig. 3e). In contrast, a transient increase in granzyme B (GrzB) levels occurred solely in the AV-infected NHPs between 9 and 15 DPI. We detected a moderate and transient peak of soluble CD40L 1 DPI in the Josiah-infected animals, whereas it occurred later, at 6 DPI, in the AV-infected NHPs. A transient release of IFNγ, peaking at 8 DPI, occurred in all infected animals, whereas the level of IL-2 increased significantly in only the Josiah-infected monkeys, until death. Finally, there was a peak in IL-5 secretion between 8 and 15 DPI only in the AV-infected NHPs.

**Innate immune responses during Lassa fever**. The number of leukocytes tended to transiently drop ~1 week after infection in all animals, but this change was not significant (Fig. 4a). The Josiah-infected NHPs showed an increase in the number of circulating granulocytes in the last days before death. Although the granulocytes were all $CD10^+$ neutrophils in the first DPI in all animals, the percentage of $CD10^-$ granulocytes steadily increased until death after Josiah infection. In the AV-infected animals, there was only a moderate and transient increase in the proportion of this cell population. The number of circulating myeloid dendritic cells (mDCs) did not change significantly during the course of the disease, except for a non-significant increase 10 and 11 DPI in AV-infected animals. No data were obtained after 11 DPI. After Josiah infection, a transient increase in monocyte count was observed 3 DPI, was followed by a drop 1 week after infection, and finally rose again in the last days before death. In AV-infected animals, the number of monocytes was elevated from 15 DPI. An activated phenotype was rapidly observed in monocytes in LASV-infected animals, with expression of CD80,

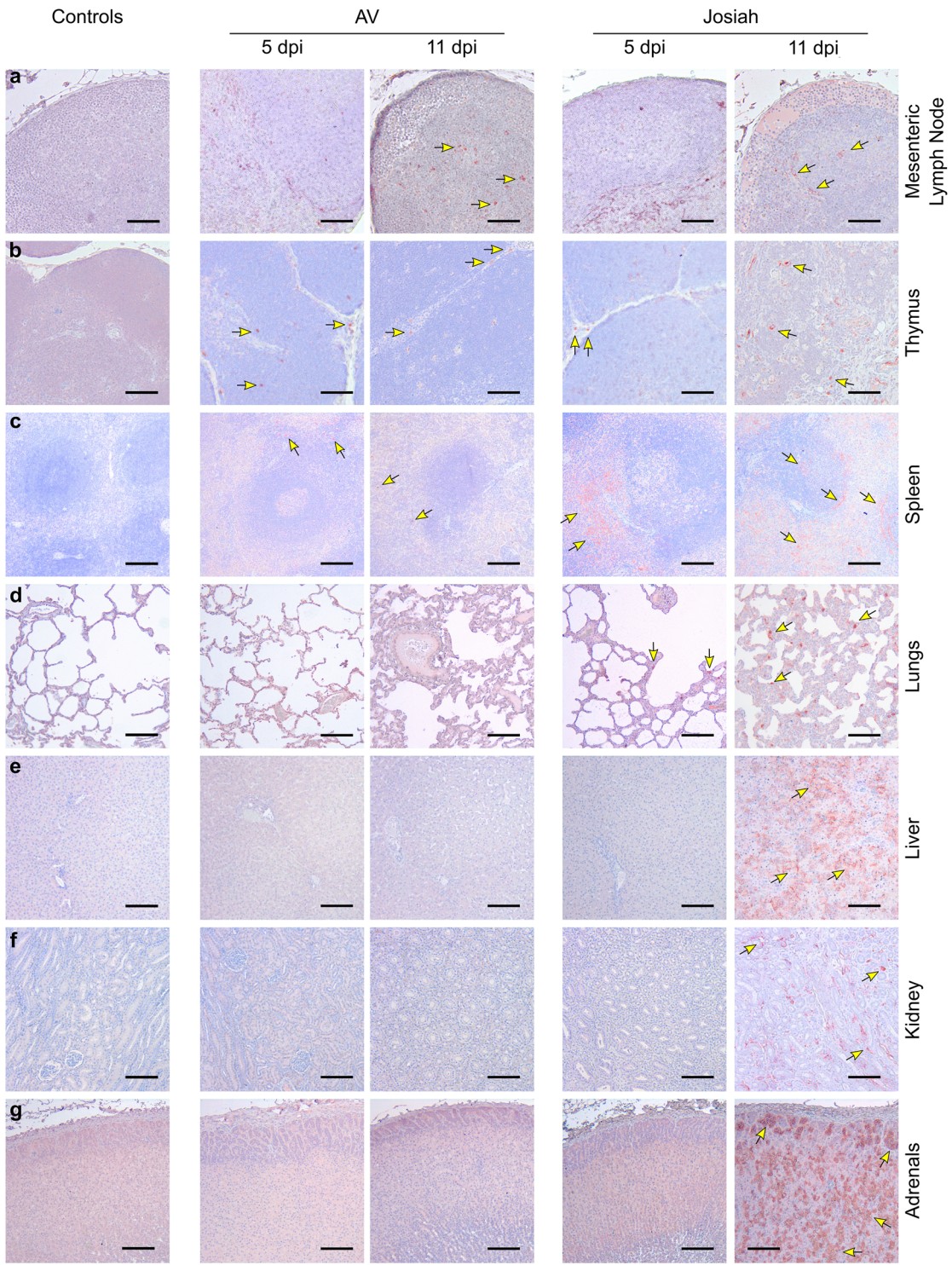

**Fig. 2 Examples of LASV spreading as a function of the severity of the disease.** Immunohistochemistry of LASV-GP (red) during the course of the disease in **a** MLNs, **b** thymus, **c** spleen, **d** lungs, **e** liver, **f** kidneys, and **g** adrenal glands. **a**, **b** Scale bars: 100 μm. **c–g** Scale bars: 200 μm. Key immunohistological features are indicated by arrows.

CD86, and CD40. The percentage of CD80$^+$ and CD40$^+$ monocytes was significantly more elevated after Josiah infection, particularly in the terminal stages. NK cell count also transiently dropped about one week after infection in all animals (Fig. 4b). An increasing percentage of NK cells proliferated at the same time in all infected animals, as demonstrated by the expression of KI67. A moderate but significant amount of NK cells expressed CD107a at their surface 6 DPI in infected animals, suggesting a

cytotoxic activity. Finally, the proportion of NKp80-expressing NK cells dramatically dropped during the 2 weeks following infection whatever the LASV strain was, whereas expression of NKG2D was not significantly modified. The percentage of KI67$^+$ NK cells was increased in spleen and MLN 11 DPI in LASV-infected NHPs, and as soon as 5 DPI in the spleen of Josiah-infected animals (Fig. 4c). A moderate but non-significant increase in GrzB$^+$ and CD107a$^+$ NK cells was measured in the

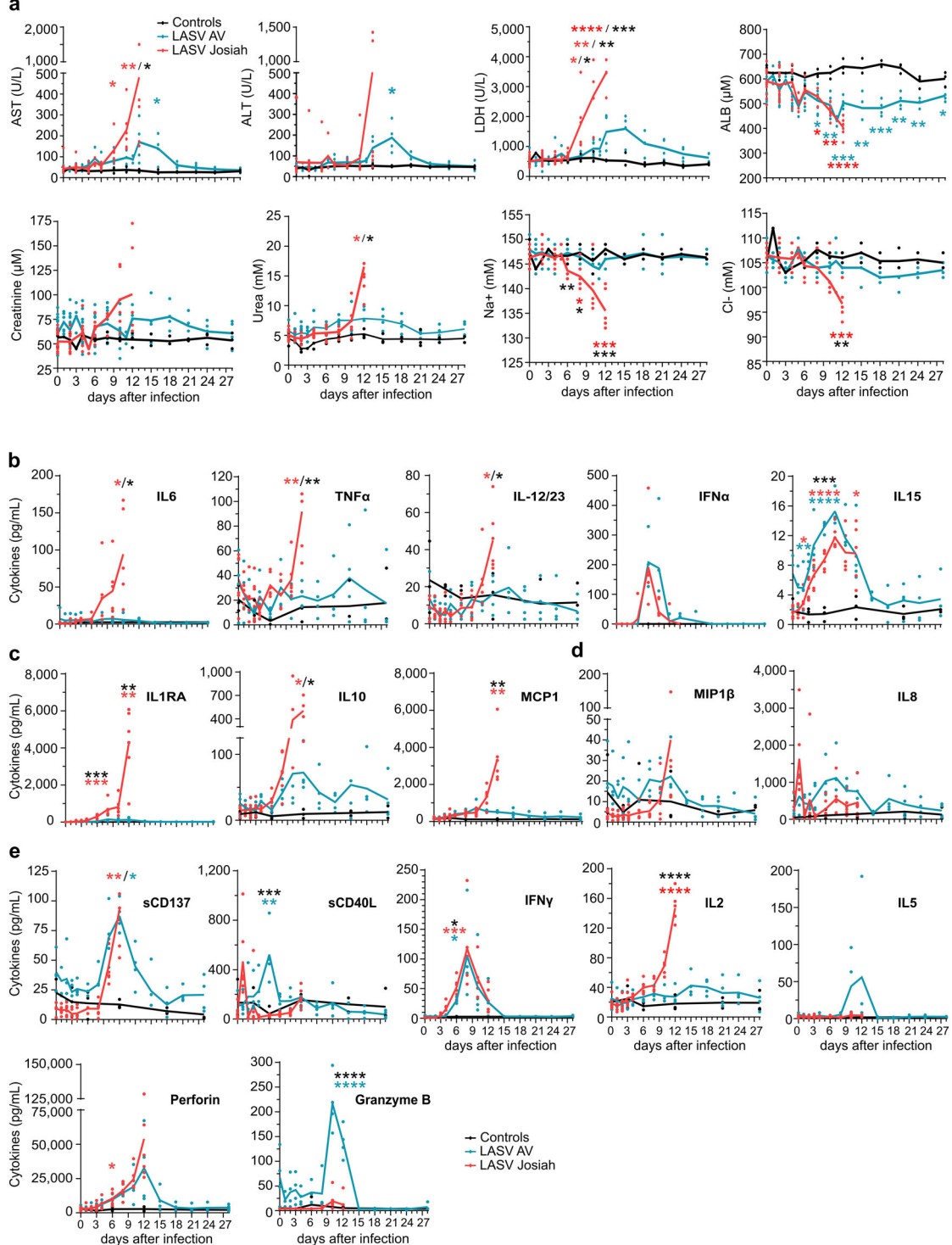

**Fig. 3 Biochemical and inflammatory responses monitoring after LASV challenge. a** Analysis of biological parameters during LASV infection. The data from longitudinally followed and sequentially killed animals at 2, 5, and 11 DPI were collected: controls (n = 3 animals), LASV-AV (n = 10), LASV-Josiah (n = 12). The individual data at the time before the death of all longitudinally followed LASV-Josiah-infected primates were collected to calculate the mean determined at 12 DPI. Results show the mean (curve) and individual data points for each group. Statistical analyses were performed and are presented as in Fig. 1. Quantification of pro-inflammatory cytokines **b**, anti-inflammatory cytokines **c**, chemokines **d**, and T-cell response-related mediators **e** in plasma according to the time after LASV infection. **b–e** Results show the mean (curve) and individual data points for each group: controls (n = 3), LASV-AV (n = 4), LASV-Josiah (n = 6). Statistical analyses were performed and are presented as in Fig. 1.

spleen of LASV-infected animals. A significantly higher proportion of GrzB⁺ NK cells was observed in AV-infected NHP MLNs. The percentage of CD107a-expressing NK cells slightly increased from 5 DPI and at 11 DPI in MLN of Josiah- and AV-infected

animals, respectively. The proportion of CXCR3⁺ NK cells transiently dropped 5 DPI in LASV-infected NHP spleen. In MLN, CXCR3⁺ NK cell percentage decreased moderately by day 5 in Josiah-infected animals and dramatically 11 DPI in AV

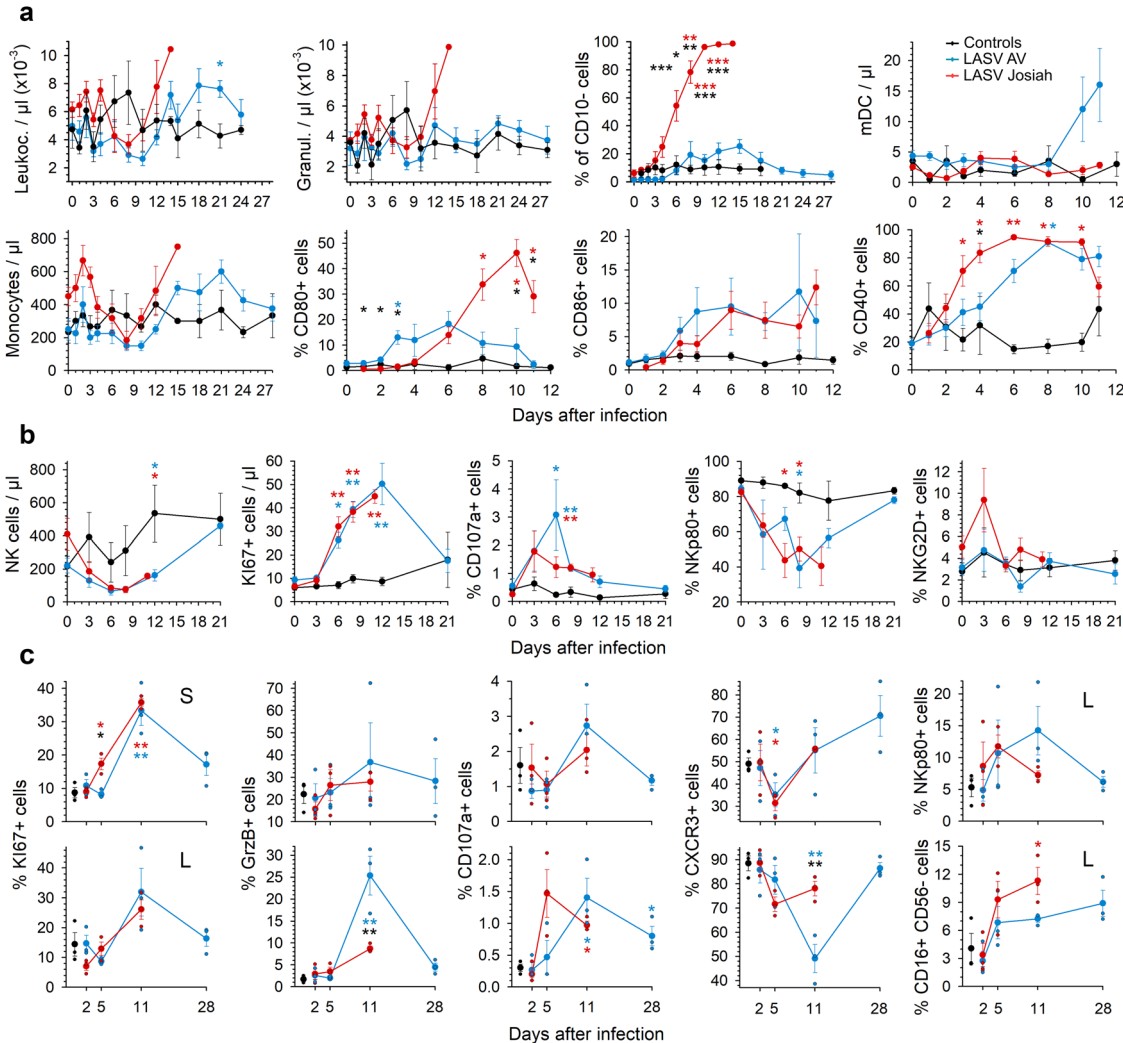

**Fig. 4 Analysis of circulating innate immune cells after LASV challenge. a** The number of leukocytes, granulocytes, mDC (HLA-DR+ CD14− CD1c+ and HLA-DR+ CD14− CD11c+), and monocytes (HLA-DR+ CD14+) in the blood is presented according to the time after LASV infection. The percentage of CD10− cells among granulocytes is also presented. The percentage of monocytes expressing CD80, CD86, or CD40 is shown. Results show the mean ± standard error of the mean (SEM) for each group: controls (n = 3), LASV-Josiah (n = 6, except for day 14 where n = 2) and AV-infected animals (n = 4) were analyzed for leukocyte, granulocyte, CD10−, and monocyte numbers. For mDC, CD80, CD86, and CD40 analysis, six AV-infected animals were analyzed from day 0 to 6 and three of them from day 8 to 11. **b** The number of circulating NK cells (CD8+ CD3− CD20− cells) is presented, as well as the percentage of KI67+, CD107a+, NKp80+, and NKG2D+ cells among NK cells (n = 3 for controls, n = 6 for LASV-Josiah, and n = 4 for AV-infected animals). Statistical analyses were performed and are presented as in Fig. 1. Individual values can be found in Supplementary data 1 for a and b. **c** The proportion of NK cells (CD8+ CD3− CD20−) expressing KI67, (granzyme B) GrzB, CD107a, CXCR3, and NKp80 was quantified in spleen (S, upper graphs) and MLN (L, lower graphs) of controls (n = 3), AV- (n = 3), and Josiah-infected (n = 3) animals, as well as the percentage of CD16+ CD56− cells among NK cells. Individual values and mean ± SEM are expressed for each group. Statistical analyses were performed and are presented as in Fig. 1.

animals. A transient increase of NKp80+ NK cells was observed 5 DPI in Josiah-infected NHPs and from 5 to 11 DPI after AV infection. However, this cell population was significantly decreased compared to control animals from 11 DPI. Finally, we observed an enrichment in CD16+ CD56− NK cells from 5 DPI, particularly in Josiah-infected animals.

**Adaptive immune response during Lassa fever.** There was a modest and transient drop in the number of B lymphocytes in both Josiah- and AV-infected NHPs ~6–8 DPI, followed by a significant increase in B-cell number from 2 weeks after infection in AV animals. There was a large increase in both LASV-specific IgM and IgG titers 8 DPI in the Josiah-infected animals, which occurred ~12 DPI in the AV-infected NHPs, with significantly higher titers 12 DPI in the Josiah-infected animals (Fig. 5b). We did not detect

neutralizing antibodies during the course of the disease, with the exception of low titers (1:20) in two of four survivors 28 DPI. There was a drop in the percentage of circulating naive/unconventional memory (N/UM) B cells (CD20+ CD38mid CD27− CD10−) ~1 week after infection, but similar changes were observed in control animals (Fig. 5c). An inverse pattern was observed for conventional memory (CM) B cells (CD20+ CD38mid CD27+ CD10−) in animals. An increase in the percentage of proliferating (KI67+) N/UM and CM B cells was observed 11 and 21 DPI in Josiah- (non-significant) and AV-infected NHPs, respectively. Finally, a transient increase in the proportion of plasma cells (CD20low CD38bright CD27+) among B cells was observed 8 and 12 DPI in Josiah- and AV-infected NHPs, respectively. In SLO, whereas the percentage of B-cell subpopulations did not change significantly, an increase of the proportion of proliferating U/UM, transitory memory (CD20+ CD38 mid CD10+ CD27−), CM, and

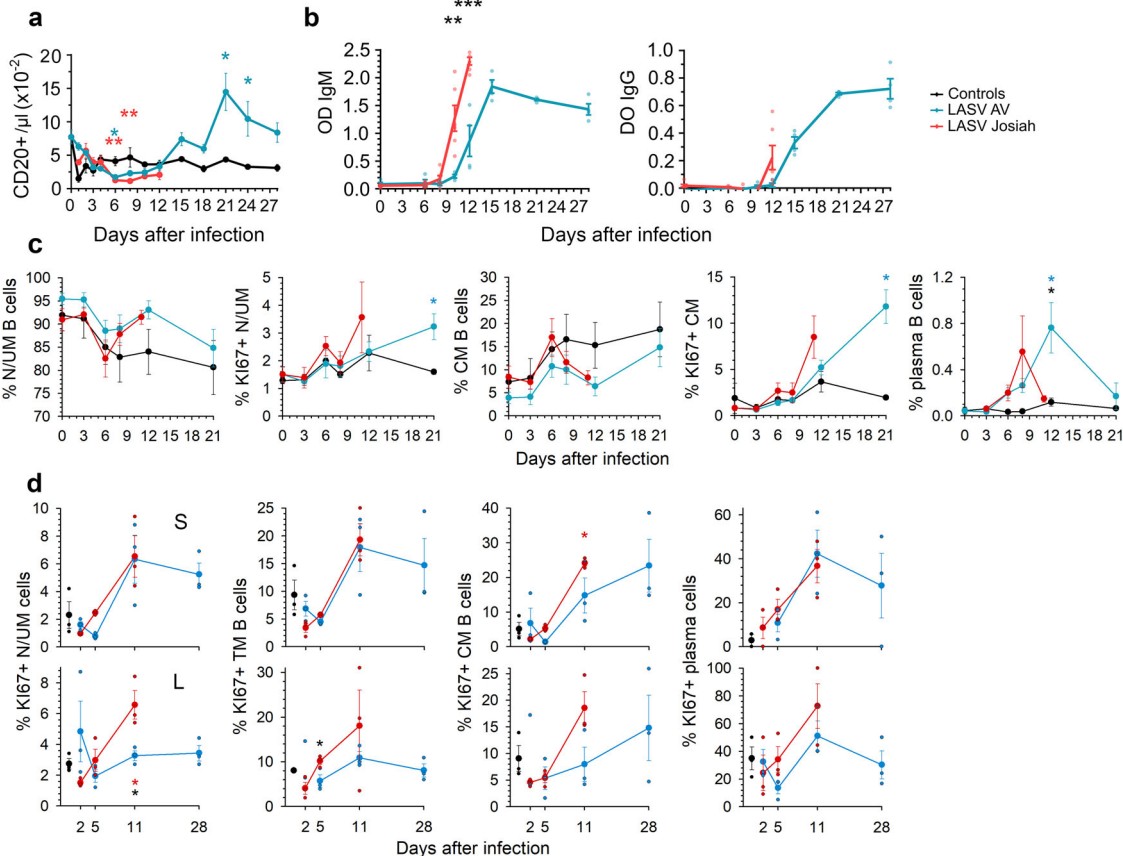

**Fig. 5 Analysis of B-cell responses after LASV challenge. a** The number of B cells (CD20⁺) in the blood is presented according to the time after LASV infection. Results show the mean ± SEM of control ($n = 3$), AV ($n = 4$), and Josiah ($n = 6$) animals. **b** Detection of LASV-specific IgM and IgG in animals after LASV infection by ELISA. Individual data (points) and mean values ± SEM for each cohort (curves) are presented. LASV-AV ($n = 4$), LASV-Josiah ($n = 6$). **$p < 0.01$, ***$p < 0.001$, as determined by a Student's $t$ test. The percentage of naive/unconventional memory (N/UM) B cells (CD20⁺ CD38$_{mid}$ CD27⁻ CD10⁻), of conventional memory (CM) B cells (CD20⁺ CD38$_{mid}$ CD27⁺ CD10⁻), and of plasma B cells (CD20$_{low}$ CD38$_{bright}$ CD27⁺) among B cells (CD20⁺) is presented, as well as the percentage of KI67⁺ cells among N/UM and CM B cells is presented for blood **c** and spleen (S, upper graphs) and MLN (L, lower graphs). Individual values can be found in Supplementary data 1 for a, b and c. **d** The percentage of KI67⁺ among transitional memory (TM) B cells (CD20⁺ CD38$_{mid}$ CD10⁺ CD27⁻) is also presented for SLOs. Results show the mean ±SEM of control ($n = 3$), AV ($n = 4$ for PBMC and three for SLOs), and Josiah ($n = 6$ and three for SLOs) animals, as well as individual values. Statistical analyses were performed and are presented as in Fig. 1.

plasma B cells was observed by 11 DPI in spleen and MLN of infected NHPs, except for N/UM B cells in MLN of AV-infected animals (Fig. 5c). Moreover, similar changes were observed in the spleen in all infected NHPs whereas B-cell proliferation was more intense in Josiah-infected animals.

A transient lymphopenia affecting CD8⁺ and CD4⁺ T cells of all infected NHPs was observed from 3 to 10 DPI (Fig. 6a). Lymphocytosis of CD8⁺ and CD4⁺ T cells then occurred only for AV-infected NHPs, starting at 12 DPI. The percentage of CD69-expressing CD8⁺ cells transiently increased after 4 DPI in all infected NHPs, whereas such an increase in CD4⁺ T cells occurred between 4 and 15 DPI solely in the AV-infected animals (Fig. 6b). The percentage of KI67⁺CD8⁺ T cells increased by 8 DPI in the AV- but not Josiah-infected animals and remained high during the course of the disease. CD4⁺ T cells showed a similar pattern, with a transient increase in the proportion of proliferative cells in the AV-infected animals from 18 to 28 DPI. The percentage of apoptotic (Annexin V⁺/7AAD⁺) CD8⁺ T and CD4⁺ T cells increased significantly between 6 and 12 DPI after Josiah infection, but not with AV. There was also an increase in the proportion of GrzB⁻ and perforin-expressing CD8⁺ T lymphocytes, starting at 6 DPI, in the AV-infected animals, whereas perforin expression of CD4⁺ T cells from AV-infected NHPs transiently increased 8 DPI. Although non-significant, the

proportion of CD8⁺ CD95⁺ cells tended to increase more and more rapidly in the Josiah- than AV-infected NHPS. Finally, only Josiah-infected NHPs exhibited a significant increase in the proportion of CD8⁺ CD279⁺ and CD4⁺CD279⁺ T cells between 6 and 10 DPI.

We assessed CD45RA, CD27, and CD28 levels to investigate the phenotypes of circulating effector/memory T cells induced during Lassa fever (Fig. S3a). The various CD8⁺ and CD4⁺ T-cell subpopulations did not change during the experiment in the control animals. There was an enrichment in naive/pre-effector (pE) 1 CD45RA⁺CD28⁺CD27⁺ (RA⁺28⁺27⁺) CD8⁺ T cells in AV-infected NHPs for all timepoints relative to the control animals, as well as a transient increase in the proportion of effector-memory (EM) 4 (RA⁻28⁺27⁻) CD8⁺ T cells. The proportion of EM4 and terminally differentiated effector-memory (EMRA) (RA⁺28⁻27⁻) CD8⁺ T cells increased between 4 and 8 DPI in the Josiah-infected NHPs. In AV-infected animals, the proportion of pE2 (RA⁺28⁻27⁺), EM2 (RA⁻28⁻27⁺), EM3 (RA⁻28⁻27⁻), and EMRA CD4⁺ T cells increased from 6 DPI. The Josiah-infected NHPs showed a transient increase in the proportion of EM4 and EMRA CD4⁺ T cells from 4 and 6 DPI, respectively.

We assessed the phenotype of T cells in the MLNs and spleens 2, 5, and 11 DPI (Fig. 7). The percentage of KI67⁺ CD8⁺ and

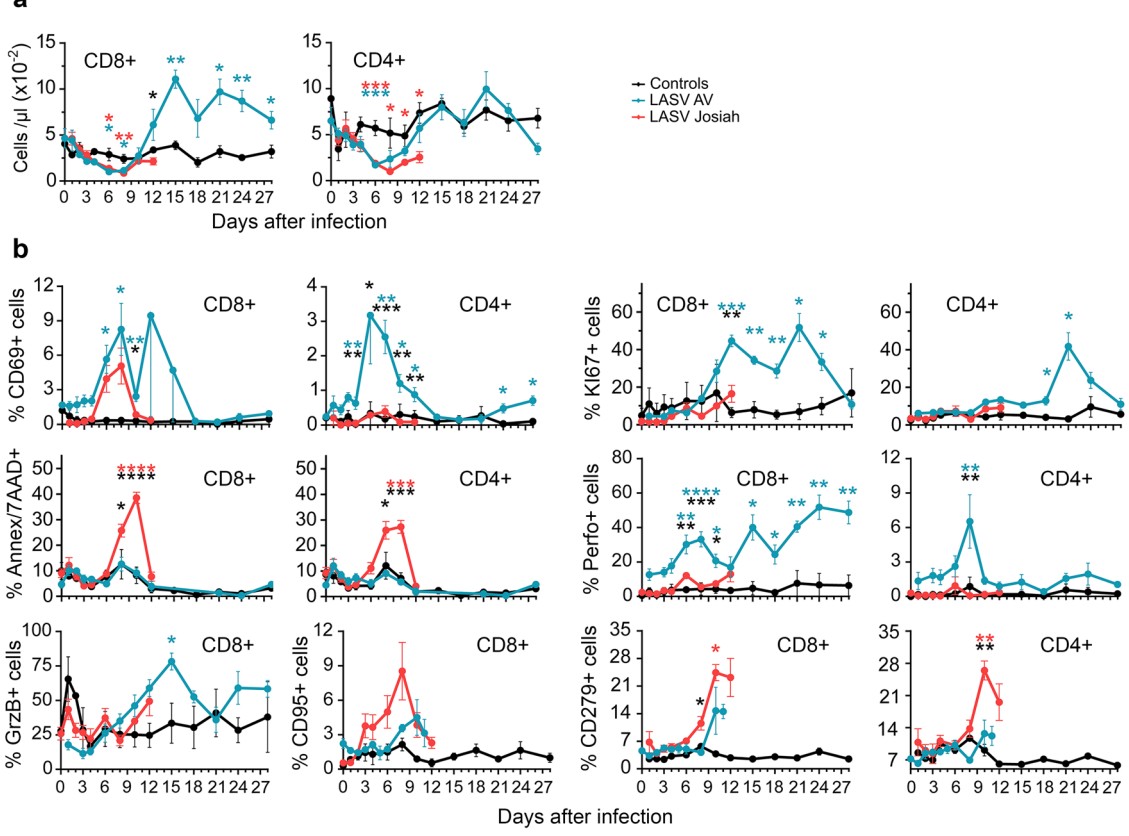

**Fig. 6 Analysis of T-cell responses in PBMC after LASV challenge. a** The number of CD8[+] and CD4[+] T cells in the blood is presented according to the time after LASV infection. **b** The percentage of circulating CD8[+] (left) and CD4[+] (right) T cells expressing CD69, KI67, Annexin V and 7AAD (Ann/7AAD), perforin (Perfo), GrzB, CD95, or CD279 is shown according to the time after LASV infection. Results show the mean ± SEM of control ($n = 3$), AV ($n = 4$), and Josiah ($n = 6$) animals. Statistical analyses were performed and are presented as in Fig. 1. Individual values can be found in Supplementary data 1.

CD4[+] T cells was significantly higher in the MLNs of AV-infected animals 11 and 28 DPI than those of mock- or Josiah-infected NHPs. In the spleen, the proportion of KI67[+] CD8[+] and CD4[+] T cells was significantly higher in all infected animals than in the control group from 11 DPI. The proportion of apoptotic CD8[+] and CD4[+] T cells was significantly higher in the MLNs of AV- than Josiah-infected NHPs 11 DPI. We observed an increase in the proportion of CD95[+] CD8[+] T cells at 11 and 28 DPI in the LNs and spleens of AV-infected animals. We also observed an increase in the proportion of CD95[+] CD4[+] T cells of AV-infected NHPs at 28 DPI. We detected a high percentage of GrzB[+] CD8[+] T cells in the spleens of all infected animals 11 DPI, while the percentage in MLNs was significantly higher in AV-infected animals. Such an observation was also made for GrzB[+] CD4[+] T cells in spleens and lymph nodes of all infected NHPs at this timepoint. Finally, we observed an increase in the percentage of perforin[+] CD8[+] T cells in the MLNs and spleens of all infected primates 11 DPI, but this raise was more important in MLNs of AV animals.

In control animals, the naive/pE1, central memory (CM)/EM1 (RA[−]28[+]27[+]) and EM4 CD8[+] T cells were the most highly represented in the spleen, MLNs, and ILNs (Fig. S3b). In AV-infected animals, there was a higher proportion of naive/pE1 and CM/EM1 T cells that further increased 11 DPI in the spleen. There were no significant differences in the CD8[+] T-cell phenotypes in the ILNs or MLNs between the AV- and mock-infected animals. In Josiah-infected monkeys, there was a higher proportion of EM1, EM3, EM4, and EMRA CD8[+] T cells 11 DPI in the ILNs and MLNs than 5 DPI, which was not the case for

AV-infected animals. Concerning the CD4[+] T-cell repertoire, the proportion of EMRA cells in the spleen was higher 5 DPI for AV-infected animals and 11 DPI for Josiah-infected macaques than in that of control animals. EM1, EM4, and EMRA CD4[+] T cells only increased in the ILNs of Josiah-infected monkeys.

**Transcriptomic changes during Lassa fever**. We analyzed the transcriptomic profiles according to infection and outcome by RNA sequencing of PBMC samples (Fig. 8). We performed pairwise comparisons to identify differentially expressed (DE) genes between mock samples and AV- and Josiah-infected NHP samples (Fig. 8a and S4). Overall, LASV infection led to significant changes in transcriptomic profiles as early as 2 DPI. These changes in expression were more marked at 4 DPI and maximal at 10 DPI. Among the 100 most DE genes were inflammation/innate immunity-related genes (*NLRC4, MNP1A, ZBED2, PRTN3, S100A8/9, MARCO, CCL23, CPA3, VGLL3, CHI3L1, PGLYRP1, LTF, RSAD2, CD1B, GRZMA, CXCL10, IFI6/27/44 L, IL27, MX1, IFIT3, ISG15, OAS1, SIGLEC1, APOBEC3A, CCL2/8,* and *XCR1*) that were upregulated, particularly in Josiah-infected animals during the terminal stages. Genes involved in extracellular matrix and cell-adhesion were also modulated, particularly during fatal Lassa fever (*NID1, TIMP3, ITGA11, TGM2, MMP8/9, OLFM4, PCDH20,* and *CADM3*), as well as some others involved in coagulation (*SERPIN, TFPI2*) and apoptosis (*CLU, BCL2L14*). The transcriptomic profiles were also highly different between the AV and Josiah infections from 2 to 10 DPI (Fig. S5). We characterized the activation of innate immunity, the cytokine response, and monocyte, T-cell, B-cell,

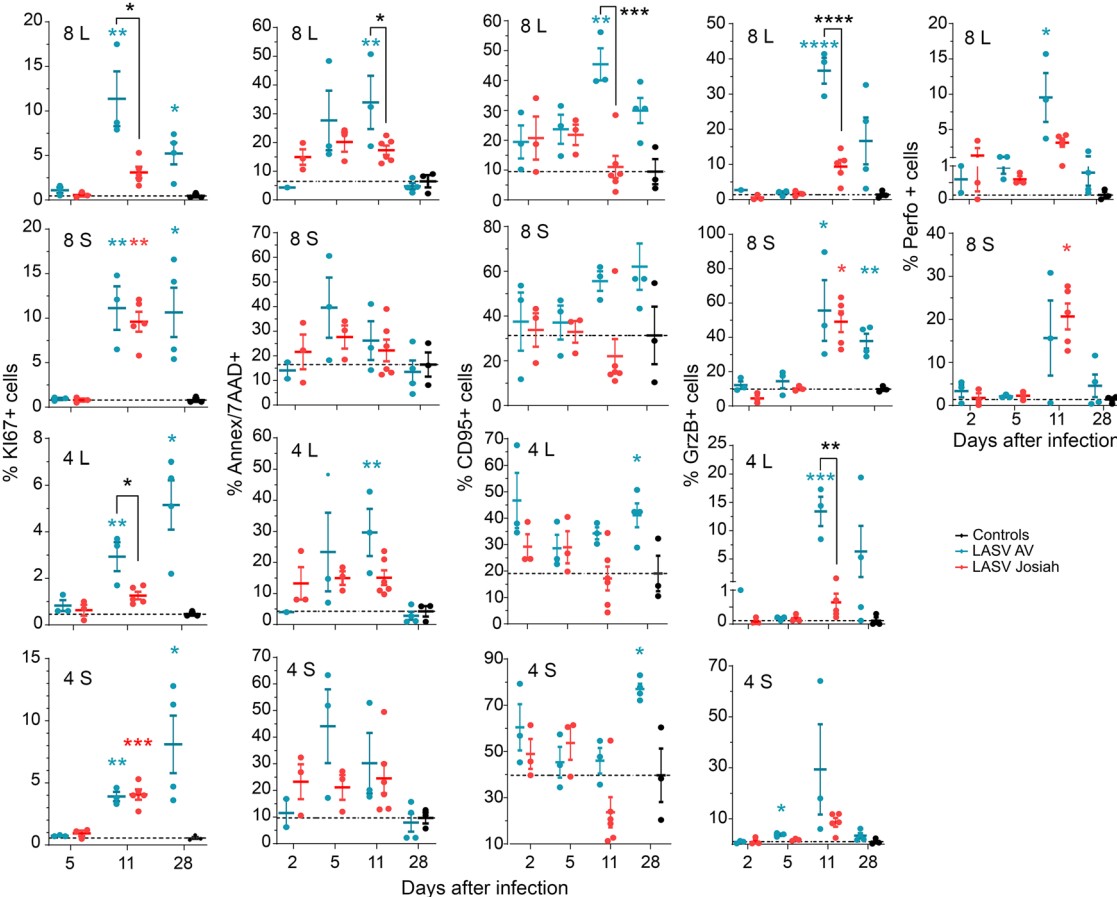

**Fig. 7 Analysis of T-cell responses in SLO.** The percentage of CD8[+] and CD4[+] T cells in MLNs (L) and spleen (S) expressing KI67, Annexin V and 7AAD (Ann/7AAD), CD95, GrzB, or perforin (Perfo) is presented in control (black dots), AV-infected (blue dots), and Josiah-infected (red dots) animals. Results show the individual data and the mean ± SEM for each group ($n = 3$). *$p < 0.05$, **$p < 0.01$, ***$p < 0.001$, ***$p < 0.0001$, as determined by a one-way ANOVA or a Kruskal–Wallis multiple comparisons test. Significant differences are presented as in Fig. 1.

and NK cell activation by analyzing the expression of genes representative of these pathways[26–28] (Fig. 8b). LASV infection induced an increase in the expression of IFN-response genes, from 2 to 10 DPI, which was maximal at day 4, when the synthesis of all IFN-response genes was upregulated. Josiah infection resulted in greater induction than AV infection. We observed similar changes for genes associated with the cytokine/chemokine response, with moderate overexpression peaking 4 DPI in AV-infected NHPs versus a dramatic increase in the synthesis of these genes 4 and 10 DPI after Josiah infection. The expression of genes related to monocyte activation was strongly upregulated after Josiah infection from 4 DPI. In contrast, the expression of T-cell activation related genes was downregulated in the same animals. B-cell-related genes were overexpressed 2 DPI and downregulated from 4 DPI in both AV- and Josiah-infected NHPs. Finally, NK cell signature was downregulated 2 and 4 DPI in Josiah-infected NHPs compared to AV and control animals. The transcription of some NK cell-related genes was induced 10 DPI in infected animals, but NK cell activation was more intense in Josiah animals compared with AV NHPs. We performed a similar analysis in the liver, spleen, and MLNs (Fig. 9). Unlike the PBMC analysis, we list only the DE genes for each pathway. IFN-response genes were upregulated in all organs after LASV infection from 2 DPI, but more intensely for Josiah-infected animals (Fig. 9a–c), except for MLNs at day 2 after AV infection (Fig. 9b). Although gene expression in the liver returned to basal levels 10 DPI in AV-infected NHPs, it remained high in the Josiah-infected animals (Fig. 9c). Certain genes related to the cytokine/

chemokine response were upregulated in spleen 4 and 10 days after LASV infection, but more particularly for Josiah (Fig. 9a). In MLNs, these genes were upregulated 4 and 10 DPI in Josiah- and AV-infected NHPs, respectively. In liver, the cytokine/chemokine response was only observed on day 10 in Josiah-infected NHPs. Monocyte-related genes were more strongly activated 4 and 10 DPI in Josiah-infected than mock- or AV-infected NHPs, whereas these genes were upregulated in both AV- and Josiah-infected animals in the MLNs. Finally, certain T-cell and/or NK cell genes related to cytotoxicity were upregulated in AV-infected animals 10 DPI in the MLNs, and to a lesser extent in spleen. Upregulation of some T/NK cell genes was also observed in the liver of Josiah-infected NHPs 10 DPI. These transcriptomic changes had major consequences on the activation of cellular pathways in both PBMC (Fig. S6) and organs (Fig. S7). An activation of type I/II IFN-response and cell cycle pathways was observed in PBMC and SLO of LASV-infected animals. In liver, a robust activation of type I/II IFN-response pathways was also observed after LASV infection. An intense downregulation of several metabolic pathways was observed in liver of Josiah-infected NHP, whereas only moderated changes were induced after AV infection. The cascade of pathophysiological events and the dysregulated inflammatory responses observed during Lassa fever are reminiscent of septic shock syndrome. We thus analyzed the expression of genes known to be DE during severe sepsis[29–35]. The pattern of upregulated (Fig. 9d) and downregulated (Fig. 9e) genes during severe sepsis was similar to that of DE genes of PBMCs obtained 4 and 10 DPI from Josiah-infected NHPs,

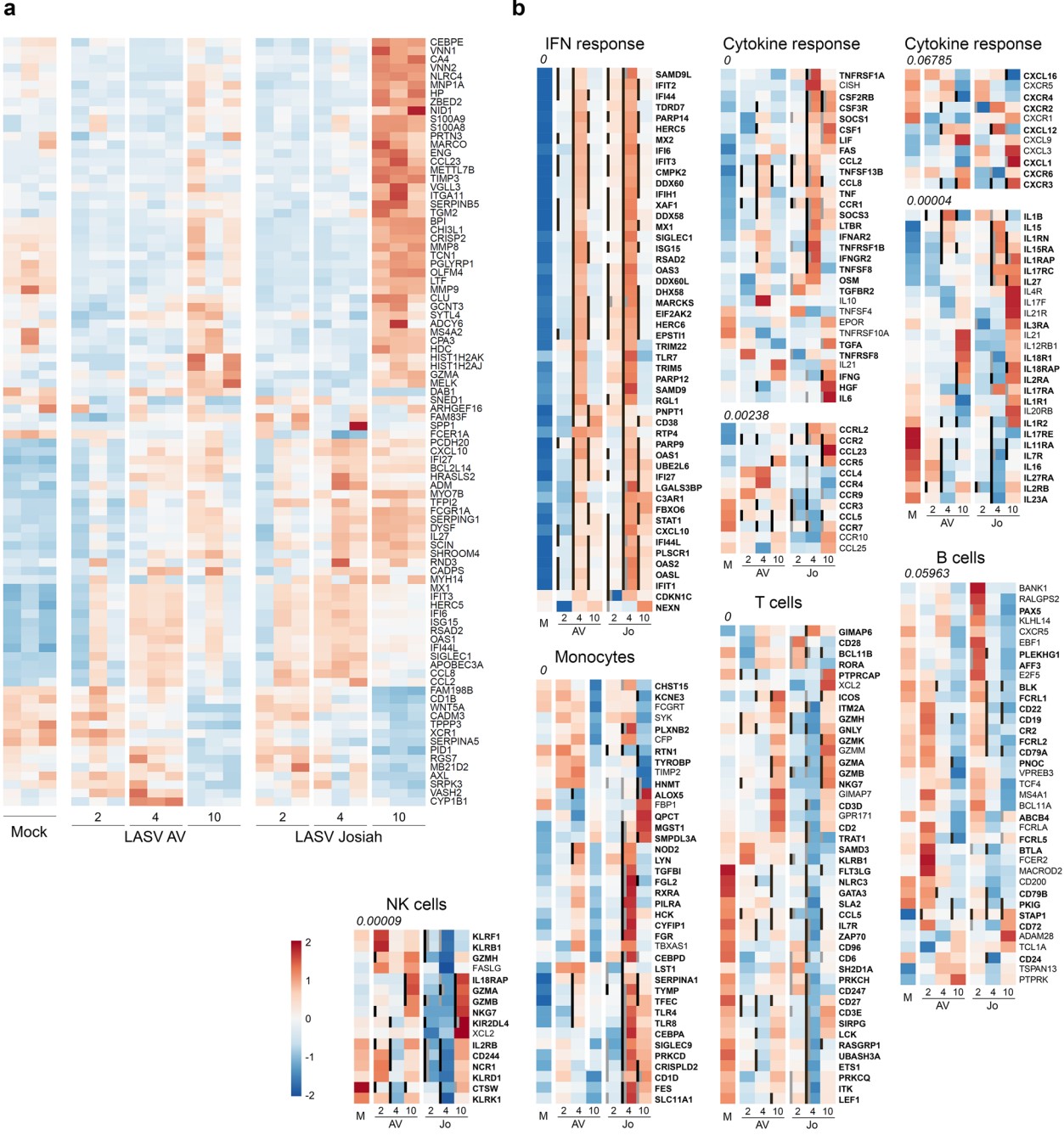

**Fig. 8 Analysis of transcriptomic data of PBMCs. a** Heatmap of the 100 most differentially expressed (DE) genes (absolute Log2 fold-change larger than 5). Gene expression was standardized using VST transformation, hence centered and scaled to make the gene expression comparable. Each column corresponds to the mean gene expression of the three animals of each group for a given timepoint. **b** Heatmap of gene expression of six gene sets. Significant genes are highlighted with bold labels and corresponding significant comparisons are displayed by gray (significant difference between Josiah and AV for a given day) and black vertical bars (significant difference with the mock condition). Gene expression was standardized using VST transformation, centered and scaled to make the gene expression comparable, hence averaged by condition and timepoint. An enrichment test was performed, using a one-tailed Fisher test. P values were adjusted on multiple comparisons using Benjamini–Hochberg correction and were presented under the gene set name.

whereas this was not true during AV infection. Among the DE genes, we selected those with the most stable expression profiles during the course of the disease and those most correlated with LASV infection or outcome, i.e., the LASV strain, to identify the set of DE genes in PBMCs that could be used as markers of LASV infection or to predict the outcome of Lassa fever. We identified seven genes allowing discrimination between mock- and LASV-infected NHPs from 2 to 10 DPI with a high probability (Fig. 10a)

and six genes able to predict the onset of Lassa fever as soon as 2 DPI (Fig. 10b). We performed RNA sequencing of PBMCs obtained 1 DPI to determine whether specific transcriptomic signatures are already present in the earliest samples and confirmed this to be the case (Fig. 10c, top). Indeed, we identified several genes for which the expression correlated with infectious status, such as *NFE2, PTP4A3, KCNG2*, and a substantial number of genes for which the expression is associated with Lassa fever

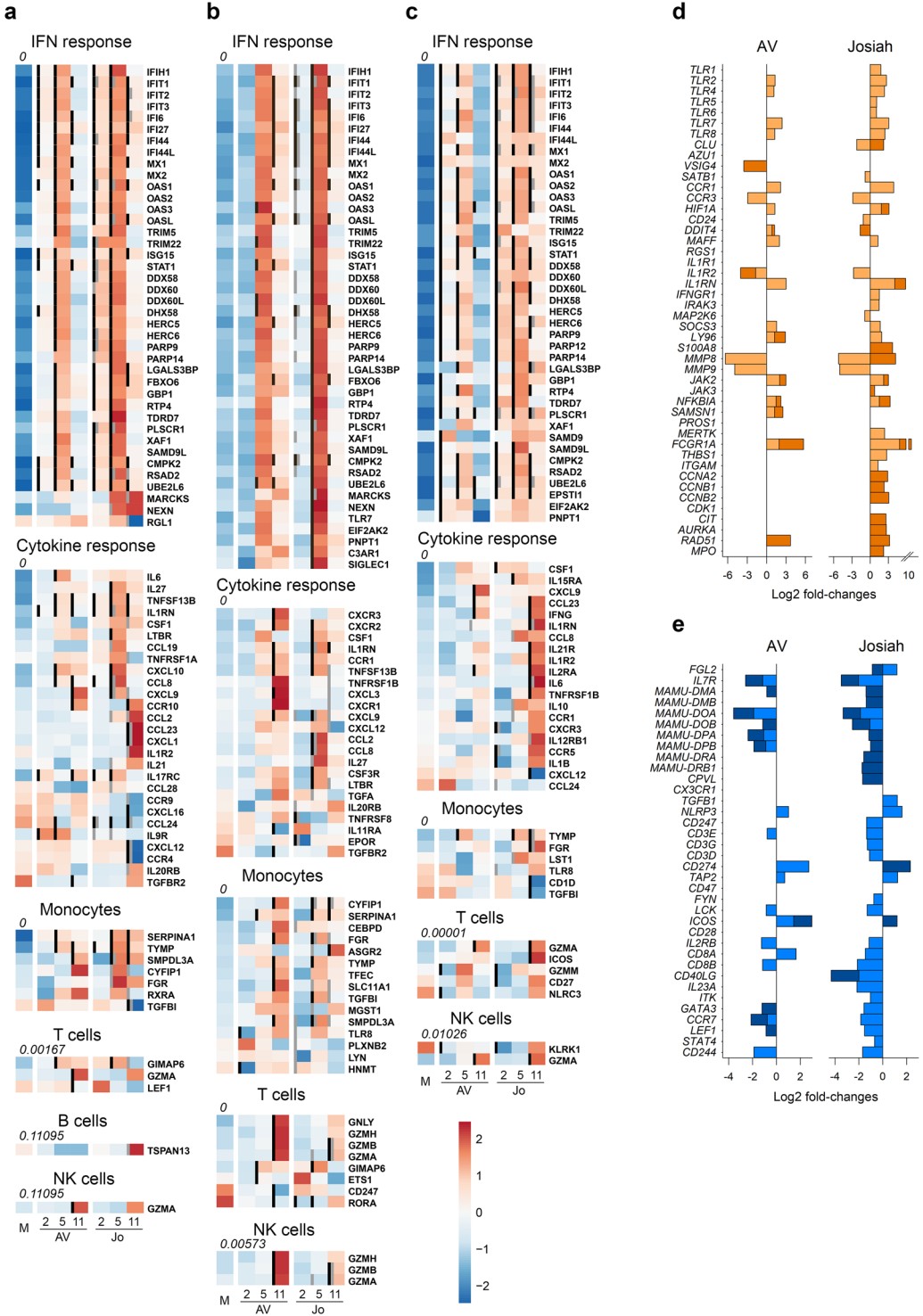

**Fig. 9 Analysis of transcriptomic data of organs and of genes related to sepsis in PBMCs.** Heatmap of gene expression of five gene sets in spleen **a** MLN, **b**, and liver **c** presented as in Fig. 8b, except that only the DE genes are listed. Gene expression was standardized using VST transformation, hence centered and scaled to make the gene expression comparable. Each column corresponds to the mean gene expression of the three animals of each group for a given timepoint. The Log2 fold-changes of genes found DE in this study and known to be upregulated **d** or downregulated **e** in PBMCs during severe sepsis were calculated between PBMCs of LASV-infected and mock animals and represented by light and dark colors for 4 and 10 DPI, respectively.

outcome. Among the set of genes defined in Fig. 10b, four appeared to already be predictive 1 DPI (Fig. 10c, bottom).

## Discussion
The cascade of pathogenic mechanisms leading to systemic shock and death during Lassa fever, as well as the immune responses

involved in the control of LASV infection, are poorly understood. Here, we describe a model of fatal and nonfatal Lassa fever in the relevant NHP model based on infection by LASV strains isolated from fatal human cases belonging to clade 4 or 5[9]. The Josiah isolate is the reference LASV strain and is known to induce a highly uniform lethal infection in this model[16,36,37]. In contrast,

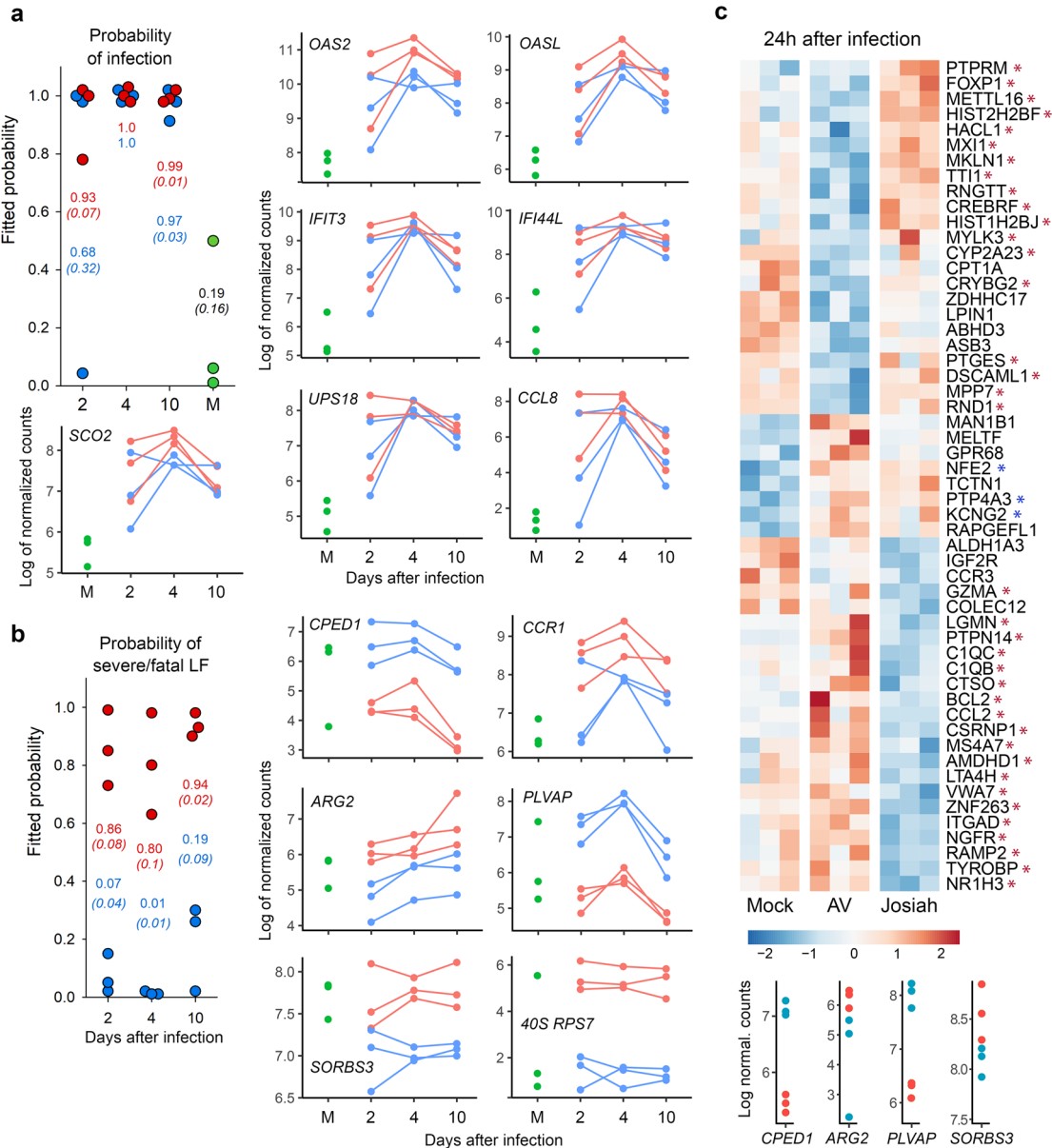

**Fig. 10 Identification of genes that can be used as early markers of infection and severity. a** Individual probabilities of infection status estimated by a random forest: each dot represents an estimation of the probability for an animal to be infected or not, at 2, 4, and 10 DPI and at 28 DPI for mock animals: the higher the probability, the greater the risk of an animal being infected. Josiah-infected animals are represented by red dots, AV-infected animals by blue dots, and mock-infected animals by green dots. The mean probability ± SEM is represented in red for Josiah, blue for AV, and black for mock animals. Probabilities were computed by applying the random forest method to the gene expression of *OAS2, OASL, IFIT3, IFI44L-201, SCO2, USP18,* and *CCL8*. The kinetics of Log2 normalized counts are also shown for each gene. **b** Individual probabilities to have a severe/fatal LASV infection were estimated and are represented as in **a**, except that mock animals were not included in this analysis. Probabilities were computed by applying the random forest method to the gene expression of *CPED1, CCR1, ARG2, PLVAP, SORBS3,* and *40 S RPS7*. **c** Heatmaps of DE genes in PBMCs at 1 DPI. The genes that were statistically associated with infection status are indicated by a blue asterisk, whereas those associated with disease severity/lethal outcome are indicated by a red asterisk. Gene expression was standardized using VST transformation, hence centered and scaled to make the gene expression comparable. Each column displays gene expression of a given animal. The expression of *CPED1, ARG2, PLVAP,* and *SORBS3* 1 DPI in PBMCs of Josiah- and AV-infected NHPs is represented at the bottom by red and blue dots, respectively. The DE of the other genes selected in **a** and **b** was not significant at 1 DPI and are therefore not represented. Transcriptomic changes measured 1 DPI could not be compared with other samples as the sequencing of these mRNA was not performed together with the others.

we previously showed that NHPs can survive infection by the AV strain[8]. Consistently, Soromba-R LASV, another strain belonging to clade 5, presented a reduced pathogenicity in macaques[17]. In the current study, all AV-infected NHPs survived and experienced clinical signs ranging from almost asymptomatic infection to quite severe Lassa fever. The clinical and biological signs observed in NHPs are reminiscent of the disease induced in

humans, with tissue and hepatic damage, illustrated by elevated AST, ALT, and LDH levels[10]. Renal failure has been recently associated with a fatal outcome, with increasing concentrations of creatinin and urea in the terminal stages of Lassa fever[10]. Albumin levels decreased thoughout Lassa fever in all infected animals and may be due to anorexia but may also have resulted from capillary leakage, abnormal fluid accumulation, liver failure, or

renal loss[38]. An electrolytic imbalance, illustrated by a drop in Na$^+$ and Cl$^-$ concentrations, was also associated with a fatal outcome. In contrast, the clinical scores remained low during infection with the AV strain and only limited alterations of biological parameters were recorded, suggesting that LASV was rapidly controlled. The first cycles of LASV replication appear to occur locally, close to the site of infection. Indeed, infectious viral particles were present in the dermis two days after SC inoculation, and the amount detected suggests active replication instead of residual inoculum. Differences in viral replication and dissemination were already evident as soon as 2 DPI, with higher viral loads in Josiah-infected NHP skin and detection of low amounts of LASV RNA in the spleen and thymus. Whether the higher replication of Josiah compared with AV was owing to increased virulence or to different host responses remains unclear. These results demonstrate, for the first time, that local replication of LASV occurs before infection of SLO and confirm that these latter represent the first organs to be targeted by LASV. Primary lymphoid organs such as thymus and BM can also be early infected by LASV. Although viremia was detected from 3 DPI in all infected animals, striking differences in viral replication and dissemination were evident very early during the course of infection between the Josiah and AV strains. Indeed, viremia was higher in Josiah- than AV-infected NHPs. Moreover, we observed greater spreading as soon as 5 DPI, with higher viral loads in the organs, with infectious viral particles detected only in lymphoid organs in AV-infected animals, whereas other organs were affected in Josiah-infected NHPs. We observed a similar pattern 11 DPI, with very high LASV titers detected in all organs in Josiah-infected NHPs. Moreover, we previously observed elevated LASV titers in oral and nasal swabs 9 DPI, confirming that the respiratory tract is a site of intense viral replication[37]. In contrast, infectious LASV was found to be mainly restricted to the lymphoid organs in all AV-infected animals, with only significantly lower amounts found in a few other organs for some. These observations were histologically confirmed, with Josiah LASV antigens present in all organs as soon as 5 DPI and massively by 11 DPI, whereas AV antigens were found only in lymphoid organs 5 DPI. The adrenals, liver, lungs, and kidneys were the organs of intense Josiah LASV replication during the terminal stages, consistent with the multiorgan failure associated with hepatic and renal dysfunction and acute respiratory distress observed for fatal Lassa fever. These results show that LASV replication is rapidly controlled and its dissemination restricted to lymphoid organs in nonfatal LF. In contrast, the lack of control leads to relentless viral replication and systemic spreading in severe/fatal LASV. The tropism of LASV for BM could play a role in the transient lymphopenia observed in infected animals, but further investigations are needed to determine if this phenomenon results from hematopoiesis defects or from peripheral deletion[39,40].

We explored the immune responses induced during Lassa fever to investigate the observed difference in viral control. Generation of memory and plasma B cells was observed in both blood and SLOs in LASV-infected NHPs, simultaneously with LASV-specific IgM and IgG, with earlier and more robust responses after Josiah infection. Genes related to the B-cell response were upregulated as soon as 2 DPI in infected animals, but the profiles were different between AV- and Josiah-infected NHPs. The expression of these genes was downregulated at 4 and 10 DPI. Together, with the lack of detection of neutralizing antibodies, these results confirm that no significant difference in the humoral responses was observed according to Lassa fever outcome and that these responses are not sufficient to control LASV infection, as previously reported[8]. In AV-infected animals, a robust CD4$^+$ and CD8+ T-cell response, involving induction and proliferation

of cytotoxic T cells, was induced in blood and SLOs by the second week after infection. The transient release of sCD137, IFNγ, IL-5, perforin, and GrzB in the plasma of AV-infected animals is also indicative of T-cell activation[41,42], as was the activation of pathways related to antigen processing and presentation in the SLOs of AV-infected NHPs. Differentiation of memory CD4$^+$ and CD8$^+$ T cells also occurred in these animals. Overall, these results are consistent with an important role of CD4$^+$ and CD8$^+$ T cells in the control of LASV infection, as previously suggested[8,43]. Importantly, both CD4$^+$ and CD8$^+$ T cells with a cytotoxic phenotype were strongly induced ~1 week after infection, suggesting that both types of T cells play a role in the lysis of infected cells[44]. The elevated amount of GrzB released is consistent with cytotoxic CD4$^+$ T-cell activation, as they appear to be more prone to secrete GrzB than CD8$^+$ T cells[42]. However, the quantification of cytokines produced by T cells after stimulation with LASV antigen-derived peptides would be necessary to analyze specific T-cell responses. Proliferating and cytotoxic CD4$^+$ and CD8$^+$ T cells were present in the spleen of Josiah-infected NHPs 11 DPI, but not in the blood or MLNs. These discrepancies could result from the functional differences between the two SLOs. Indeed, blood-borne antigens directly enter the spleen, whereas APCs bring antigens to the LNs through the lymphatics and the elevated viremia present during Josiah infection may explain the activation of splenic T cells[45]. The release of IFNγ, sCD137, and IL-2 suggests T-cell activation in these animals, although IL-2 could also come from DCs activated by pathogen-associated molecular patterns (PAMPs) such as viral RNA[46]. Changes in memory T-cell subpopulations were also observed in the blood and SLOs of Josiah-infected animals, but their overall T-cell responses appear to be defective. Indeed, high proportions of apoptotic and CD279$^+$ CD4$^+$ and CD8$^+$ T cells and CD95$^+$ CD8$^+$ T cells were detected in the blood. These observations suggest that massive T-cell death and exhaustion were induced during Josiah infection[47]. Consistently, the induction of CD279$^+$ T cells has been associated with severe Ebola virus disease[48]. The transcriptomic results in PBMCs and SLOs are also consistent with adaptive T-cell responses of lower intensity in the Josiah-infected NHPs. Finally, pathways linked to mitosis and cell cycle were activated in SLOs of infected NHPs, but more intensively after AV infection, probably reflecting a more robust adaptive immunity in these animals.

NK cells are probably involved in the innate response against LASV, as suggested by their proliferation and acquisition of cytotoxic markers as well as their transcriptomic signatures in blood and SLOs of infected animals. The enrichment in CD16$^+$ CD56$^-$, NKp80$^+$, GrzB$^+$, and CD107a$^+$ cells in MLN and spleen for the latter is consistent with the induction of cytotoxic NK cells in SLOs[49,50]. The NKp80$^-$ NK cells that circulated in LASV-infected NHPs could result from relocalization of NKp80$^+$ cells into tissues or to downregulation of NKp80 expression after exposure to monokines[51]. The downregulation of CXCR3 expression in NK cells in SLOs is probably owing to ligand-induced internalization. Indeed, CXCL9, CXCL10, and CXCL11 are the ligand for CXCR3 and attract NK cells into inflamed tissues[52]. The synthesis of CXCL9 and CXCL11 mRNA observed in blood and SLO of LASV-infected animals is consistent with this hypothesis. Finally, NK cells could be the source of the IFNγ released in plasma in infected animals. Although further investigations are needed to understand the role played by NK cells, a similar response observed in both Josiah- and AV-infected animals argues against a primary role of NK cells in Lassa fever outcome.

A balanced and self-limiting inflammatory response was induced during AV LASV infection. IFN-response gene synthesis was only induced 4 to 5 DPI in the PBMCs and organs while

cytokine/chemokine and monocyte-associated genes were only moderately overexpressed. These observations were mirrored by the moderate and transient release of inflammatory/anti-inflammatory cytokines and chemokines into the plasma. This balanced inflammatory response may have played an important role in the induction of efficient T-cell immunity, as these cytokines are known to serve as costimulatory signals for T-cell activation[53]. In sharp contrast to the response in AV-infected NHPs, an overwhelming, dysregulated, and potentially deleterious inflammatory response appeared to be induced during Josiah infection. The expression of IFN-response genes was upregulated as soon as 2 DPI in the PBMCs and organs up to death, and the levels were very high from 4 to 5 DPI. This transcriptomic response can be, at least within PBMCs, attributed to monocytes, given the intense overexpression of monocyte-related genes observed at the same time. Consistently, a large proportion of circulating monocytes was activated in the blood of Josiah-infected animals during the disease, whereas this was more balanced after AV infection. Similar results in the transcriptomic response of PBMC have been previously reported in a fatal model of Lassa fever in macaques after aerosol infection, with early induction of IFN-responsive genes and TLR signaling pathways[54], and also in the Lassa fever-like model of intravenous LCMV infection of macaques[20]. Genes related to cytokine/chemokine responses were robustly expressed during Josiah infection in both PBMCs and organs. In the liver, these transcriptomic changes were maximal during the terminal stages, which is consistent with the liver alterations observed at the same time. The liver metabolism was highly suppressed during Josiah infection, probably contributing to liver failure and pathogenesis. The metabolic pathways were only moderately affected after AV infection, confirming that liver function was preserved during nonfatal Lassa fever. Very high concentrations of inflammatory/anti-inflammatory cytokines and chemokines were also detected in the circulation, with levels steadily increasing until death. These mediators may be released by activated monocyte/macrophages, which would be consistent with the activated phenotype of circulating monocytes observed in Josiah-infected NHPs as well as the gene signature of activated macrophages in their SLOs. This unbalanced and excessive inflammatory response may be involved in the defective T-cell activation observed in the SLOs. Indeed, the homing and motility of T cells within the SLOs and their contact with APCs are dictated by the complex and well-regulated expression of homeostatic chemokines by stromal cells of the SLOs[45,55–57]. Dysregulated chemokine synthesis may have altered the traffic of naive T cells within the SLOs, preventing the induction of effective immunity. It has been previously shown that arenavirus tropism, including that of LASV, for fibroblastic reticular cells may have a deleterious effect on the architecture of the SLOs and consequently on the induction of the immune response[58–60]. Consistent with this finding, a loss in lymphoid follicle architecture was evident in LASV-infected NHPs. The large number of CD10− granulocytes, a phenotype characteristic of immature myeloid cells, that circulated in the final days before death in Josiah-infected NHPs may have also been involved in the defective T-cell response[61]. Indeed, these cells circulate during severe sepsis and cancer and are related to myeloid-derived suppressor cells. They may exhibit T-cell suppressive properties and their presence is associated with a poor prognosis in sepsis. Furthermore, infection of DCs has been observed within the SLOs of Josiah-infected NHPs 7 DPI[16]. As LASV-infected DCs fail to induce T-cell responses[62,63], this tropism could also have a role in the lack of T-cell activation in Josiah-infected NHPs. Finally, the large proportion of apoptotic T cells within the PBMCs and SLOs probably also contributed to immunosuppression. However, further studies would be needed to decipher the mechanisms involved in the defective T-cell immunity following Josiah LASV infection. Overall, the observations made during Josiah infection are reminiscent of septic shock syndrome. Indeed, the cytokine/chemokine storm, combining inflammatory and anti-inflammatory compounds, detected in the plasma at the late stage is similar to that associated with severe sepsis and septic shock[64]. Very high levels of IL-10 were released in the circulation of Josiah-infected animals. This mediator is probably not produced by PBMC, with regards to their low IL-10 mRNA levels, but rather in tissues/organs. Consistently, the synthesis of IL-10 mRNA was significantly increased as soon as 4 DPI in the liver of Josiah-infected monkeys. The massive production of mRNAs coding for these mediators was also detected in PBMCs and organs from 4 to 5 DPI until death and pathways related to TLR- and RIG-like receptors (RLR), Janus kinases, mitogen-activated protein kinases, signal transducers and activators of transcription (STAT), and NF-κB were activated as observed during severe sepsis[65]. Moreover, the most altered pathways in PBMC, SLOs, and liver were those related to type I and type I IFN responses, with dramatic activation in infected animals, and more particularly in Josiah-infected NHPs. These observations are consistent with the soluble mediators detected in plasma and with a septic-like syndrome. Most clinical and biological signs observed during Josiah infection also reflect those of severe sepsis: lymphopenia and T-cell apoptosis, thrombocytopenia, liver failure with hepatic necrosis, renal failure, vascular permeability, edema, and acute respiratory distress syndrome (ARDS)[65–69]. The dramatic thickening of the alveolar walls and interstitial pneumonitis observed in the lungs of Josiah-infected animals are consistent with ARDS. The dramatic Josiah LASV replication in the lungs could have favored an exacerbated local inflammatory response, contributing to the pathology observed in these tissues. Furthermore, large sets of genes related to the inflammatory response and T-cell immunity, known to be up and downregulated, respectively, in correlation with the severity of sepsis, were similarly modulated in Josiah-infected animals[29–35], further suggesting that highly similar pathophysiological mechanisms are recruited during sepsis and Lassa fever.

The outcome of Lassa fever appears to be determined very early. Indeed, as soon as 1 DPI, we were able to identify DE genes in PBMCs associated with LASV infection and Lassa fever outcome. At a time when LASV replication is still restricted to the local inoculation site, a systemic PBMC response can already be observed, as previously reported[19]. Moreover, upregulation of IFN-response-related genes was also evident 2 DPI in the spleen and liver of all LASV-infected NHPs relative to control animals and in the spleen, LN, and liver of Josiah-infected animals compared to the other NHPs. This observation confirms that a systemic response was induced very early after LASV infection, at a time when the infection was clinically silent. From 2 to 10 DPI, certain IFN-response genes were upregulated in LASV-infected NHPs relative to control animals and could be used as high probability markers of infection, as proposed by other authors[19,70]. Similarly, another set of genes was DE between AV- and Josiah-infected NHPs throughout the course of infection, allowing prediction of the outcome of Lassa fever from the early incubation period to terminal stages, as proposed for other acute viral infections such as dengue[71]. The analysis of the expression of these genes by RT-PCR could be used for diagnostic purposes, as previously proposed for other genes[54], assuming that they will likely not be specific for Lassa fever but rather represent an acute viral infection signature. Nevertheless, during an outbreak, such an approach would allow determining whether a subject has been infected very early after a high-risk contact and provide an idea of the severity of the coming disease. Whether the same sets of genes are modulated during human Lassa fever is yet to be determined,

but similar early transcriptomic profiles are probably induced in the PBMCs of patients. The relative specificity of these gene sets for Lassa fever could also be challenged against other infectious diseases. The prognostic of Lassa fever severity could be useful in epidemic setting when intensive care can be provided to a limited number of patients. Our results suggest that the orientation toward an efficient or defective immune response after LASV infection is decided in the first hours after infection. They allow us to propose the following hypothesis to explain the pathophysiogenesis of Lassa fever. The first cycles of viral replication occur at the site of inoculation. Then LASV reaches the SLOs, which probably serve as a viral reservoir before systemic infection. In nonfatal Lassa fever, a balanced inflammatory response is induced early after infection and an efficient T-cell response is generated. Viral replication is restricted to the SLOs and no systemic dissemination of LASV occurs, probably owing to an efficient innate and/or T-cell response. This lack of systemic spreading avoids the appearance of severe clinical signs and only asthenia, anorexia, and fever are observed. From 12 DPI, viremia decreases concomitantly with the appearance of LASV-specific IgG and a robust $CD4^+$ and $CD8^+$ T-cell response. This balanced humoral and cellular response probably allows for complete control of LASV and recovery. In fatal Lassa fever, LASV is not restricted to the SLOs and systemic viral spreading occurs, with intense viral replication in virtually all organs and tissues, despite a more intense induction of the IFN-response in the early stages, possibly owing to higher viral replication. This massive replication generates substantial amounts of PAMPs, as double-stranded RNA intermediates or single-stranded RNA, which intensively activate cells through TLRs and RLRs, inducing a cytokine/chemokine storm, a defective T-cell response, and a pathological cascade similar to that induced during septic shock syndrome. Multiorgan failure then occurs, followed by death in a hypovolemic, hypoxic, and hypotensive context. Although the early events that dictate the evolution towards LASV control or severe HF and death remain unclear, this study brings new insights into the pathogenesis of Lassa fever and identifies transcriptomic signatures that could serve as an early marker of LASV infection and of Lassa fever prognosis.

## Methods

**Cell cultures**. Vero E6 cells were grown in Glutamax Dulbecco Modified Eagle's Medium (DMEM, Life Technologies) supplemented with 5% fetal bovine serum (FBS) and 0.5% penicillin–streptomycin (P/S).

**Viruses**. LASV-AV (GenBank FR832711.1 and FR832710.1) and LASV-Josiah (GenBank HQ688674.1 and HQ688672.1) strains were kindly provided by Dr Stephan Becker (Philipps-Universität Marburg, Marburg, Germany). AV and Josiah strains were passaged four and five times before generating a stock, respectively. Stocks were produced on Vero E6 cells at a multiplicity of infection (MOI) of 0.01 in DMEM supplemented with 2% FBS and 0.5% P/S. Culture supernatants were harvested, titrated on Vero E6 cells, checked for the absence of *Mycoplasma spp.*, aliquoted, and frozen at −80 °C.

**Infection of animals**. All infectious work with LASV and sample inactivation was performed in a maximum containment laboratory (Laboratoire P4 Jean Mérieux, Lyon, France). All cynomolgus monkeys (*Macaca fascicularis*) (12 females, 16 males, age range 2–3 years, weight range 2–5 kg) were purchased from SILABE (Simian Laboratory Europe, Niederhausbergen, France). One group of four and one group of six monkeys were infected with a subcutaneous injection (at the back of the thigh) of 0.5 mL PBS containing $10^3$ FFU of LASV-AV or LASV-Josiah strain, respectively. Another control group of three monkeys was injected subcutaneously with 0.5 mL PBS. Animals were followed for clinical signs of the disease and euthanized according to a scoring based on body temperature, body weight, feeding, hydration, behavior, and clinical signs. The experimentation endpoint was placed at 28 days post-challenge and all animals that had survived to this point were killed according to validated experimental procedures. Blood draws were performed daily from the day of infection to the fourth DPI, every 2 days from the fourth day to the 12th DPI, and every 3–4 days between the 12th day and the end of the experiment. BM was harvested on the second, fourth, seventh, and

tenth day, as well as during necropsies. In parallel with the longitudinal follow-up of animals, groups of three monkeys infected with $10^3$ FFU of LASV-AV were sequentially killed 2, 5, and 11 DPI. Other groups of three monkeys were also infected with $10^3$ FFU of LASV-Josiah and sequentially killed 2 and 5 DPI. Blood and BM samples were harvested using the same protocol as described above. Full necropsies were performed on each animal and organs were harvested for virologic and pathological analyses. All procedures were approved by the Comité Régional d'Ethique pour l'Expérimentation Animale Rhône Alpes (file number 2015062410456662, CECAPP, UMS3444/US8, Lyon, France).

**Hematology and biochemistry analyses**. Hematology was completed on a MS9 (Melet Schloesing Laboratories, Osny, France) and the following parameters were evaluated in blood samples harvested in ethylenediamine tetraacetic acid (EDTA) tubes: red blood cell count, hemoglobin concentration, hematocrit, mean corpuscular volume, mean corpuscular hemoglobin, mean corpuscular hemoglobin concentration, platelets count, neutrophil count, lymphocyte count, monocyte count, eosinophil count, and basophil count. Part of the biochemistry analyses was completed on a Horiba Pentra C200 (Horiba, Kyoto, Japan) and the following parameters were evaluated in blood samples harvested in heparin tubes: ALT, AST, creatinine kinase, CRP, total bilirubin, direct bilirubin, gamma glutamyltransferase, LDH, albumin, total protein, iron, uric acid, and calcium. The other part of the biochemistry analyses was completed on an i-STAT (Abbott, Princeton, USA), with an evaluation of the following parameters: sodium, potassium, chloride, total carbon dioxide, glucose, anionic gap, creatinine, and urea.

**LASV RNA detection**. Viral RNA was extracted from plasma, cerebrospinal fluid, and urine using QIAamp Viral RNA Minikits (Qiagen, Hilden, Germany), according to the manufacturer's instructions. Organs were treated with RNA later (Qiagen) prior to freezing. After weighing and grinding the various tissues, total RNA was extracted using ViralRNeasy Minikits (Qiagen), according to the manufacturer's instructions. Quantification of viral RNA was performed using the EuroBioGreen Lo-ROX qPCR mix (Eurobio, Les Ulis, France), with primers specific for LASV *NP* sequence: 5′- CTTTCACCAGGGGTGTCT-3′ and 5′-GTCACC TCAGACAATGGATGG-3′ for LASV-Josiah and 5′-CTCTCACCCGGAGTAT CT-3′ and 5′-CCTCAATCAATGGATGGC-3′ for LASV-AV. RT-PCR was performed on a Light Cycler 480 II (Roche, Boulogne-Billancourt, France).

**LASV titration**. Vero E6 cells were infected with sequential dilutions of body fluids and resuspended tissue and incubated at 37 °C in 5% $CO_2$ for 7 days with carboxymethylcellulose (CMC) (1.6%) (BDH Laboratory Supplies, Poole, UK) in DMEM supplemented with 2% FBS. Infectious foci were detected by incubation with monoclonal antibodies (mAbs) directed against LASV antigens (mAbs L52-54-6A, L53-237-5, and YQB06-AE05, generously provided by P. Jahrling, USAMRIID, Fort Detrick, MD), followed by PA-conjugated goat polyclonal antimouse IgG (Sigma-Aldrich).

**Quantification of cytokines in plasma**. Thirty-seven analytes were measured in plasma samples using the following multiplex magnetic bead assays: NHP cytokine/Chemokine Magnetic Bead Panels® I (GM-CSF, TGFα, GCSF, IFNγ, IL-2, IL-10, IL-15, sCD40L, IL-17, IL-1ra, IL-13, IL-1β, IL-4, IL-5, IL-6, IL-8, MIP-1α, MCP-1, TNFα, MIP-1β, IL12/23p40, VEGF, and IL-18) and II (FGF-2, Eotaxin, IL-16, fractalkine, sCD137, granzyme A and B, IL-1α, IL-23, IP-10, sFasL, TNF-β, IL-28A, and perforin) (Merck). Plates were prepared according to the manufacturers' recommendations and analyzed on a Magpix® instrument (Merck). IFN-α2 levels were detected in the plasma of animals using matched pair antibodies and standard protein (Life technologies), according to the manufacturer's instructions. For the detection of LASV-specific IgG, polysorp plates (Nunc) were coated overnight with lysates of LASV− or mock-infected Vero E6 cells as positive and negative antigens, respectively. Plasma samples were then incubated for 2 h on both antigens at dilutions of 1:250, 1:1000, 1:4000, and 1:16,000. Plates were further incubated with polyclonal peroxidase-conjugated antibodies directed against NHP IgGγ-chain (Kirkegaard and Perry Laboratories). Tetramethylbenzidine (TMB) (Eurobio) and $H_3PO_4$ solutions were used for detection and the optical densities (OD) measured. For the detection of LASV-specific IgM antibody titers, Maxisorp plates (Nunc) were coated overnight with monkey anti-μ antibodies at a concentration of 5 μg/ mL and then saturated for 2 h with a 2.5% PBS-SAB solution before adding diluted plasma samples at dilutions of 1:100, 1:400, and 1:1600. Negative and positive LASV antigens were then added to the plates for 1 h before incubation with a mouse anti-LASV antibody cocktail. Plates were then incubated with anti-mouse peroxidase-conjugated antibodies before revelation with TMB and $H_3PO_4$ and measuring the OD.

**Neutralization assays**. For plaque reduction neutralization assays, plasma samples were diluted and mixed with 150 FFU of LASV strain Josiah. After 1 h, the mixtures were added to Vero E6 cells and incubated for 1 h before the addition of CMC diluted in DMEM. After seven days, the number of foci was calculated by focus-forming immunodetection using anti-LASV antibodies.

**Histology**. Histopathology and immunohistochemistry were performed on macaque tissues. Following fixation and inactivation of tissues for a minimum of 14 days in 10% neutral-buffered formalin and one formalin exchange, tissues were removed from the BSL-4 and processed using a STP120 (Microm Microtech, Brignais, France) and embedded with paraffin using aTES99 device (Tech-Inter, Thoiry, France). Samples were sectioned (3–5 μm) and the resulting slides stained with hematoxylin and eosin. Anti-LASV virus immunoreactivity was detected using a custom anti-mouse GP2c antibody at a 1:50 dilution after unmasking of the paraffin embedded samples. The secondary antibody was a ready-to-use peroxidase coupled anti-mouse (N-Histofine, Microm Microtech). Samples were then counterstained with hematoxylin before observation with a Leica DMi8 microscope and LASX software.

**Flow cytometry**. Blood cell populations, proliferation, and cytotoxic activity during the course of the disease were quantified from 50 μL of fresh whole blood. Surface antibodies to human or NHP CD3, CD4, CD8, CD10, CD20, CD27, CD28, CD45RA, CD56, CD69, CD80, and CD95 from BD Biosciences, CD86, CD1c, HLA-DR, CD14 from Miltenyi Bitotech, and CD40 and CD279 from BioLegend were stained for 30 min on ice. For surface staining only, cells were fixed after lysing of the red blood cells using the Immunoprep Reagent System (Beckman Coulter) according to manufacturer's instructions.

For intracellular staining, red blood cells were lysed using PharmLyse (BD Biosciences) and white cells were stained with surface antibodies, permeabilized using the FoxP3 staining buffer set (Miltenyi), and stained using antibodies to Ki67, GrzB (BD Biosciences), and perforin (Mabtech) before fixation in PBS with 1% paraformaldehyde (PFA). Apoptotic and necrotic cells were identified using the Annexin V Apoptosis Detection Kit with 7AAD (BioLegend) according to the manufacturer's instructions. Spleens, MLNs, and ILNs were manually crushed and the tissue filtered with a 70-μm cell strainer to recover the cells. Red blood cells were then lysed using ACK buffer (ThermoFisher Scientific) and cells from the organs suspended in fresh RPMI medium supplemented with 1% Hepes, 5% fetal bovine serum, and 0.5% P/S. Cells were then directly stained for intracellular and extracellular markers using the same protocol and reagents described above. For NK and B cells analysis, frozen PBMC and organs were thawed and washed in RPMI 5% FCS, 1% Hepes, non-essential amino acids and 0.5% P/S. Cells were then stained with surface antibodies for 30 min on ice with CD27, CD20, CD107a, and CXCR3 antibodies (BD Biosciences), CD16 (Biolegend), NKp80 and NKG2d (Miltenyi Biotech), and CD38 (Stemcell). Other antibodies used were described above. Cells were washed once in PBS, 0.5% FCS and 2 mM EDTA before fixation with 1% PFA. Intracellular staining with Ki67 was performed using the same intracellular staining method than above. Cells were analyzed using a 10-color Gallios cytometer (Beckman Coulter) and the data analyzed using Kaluza software (Beckman Coulter).

**Transcriptomic analysis**. Total RNA was extracted from PBMCs, splenocytes, MLN, and liver using the RNeasy Extraction Kit (QIAGEN). RNA samples were then quantified using the Quantifluor RNA system (Promega) and qualified using Pico RNA Chips on a Bioanalyzer 2100 (Agilent). External RNA Controls Consortium (ERCC) RNA Spike-in Mix 1 (ThermoFisher Scientific) was added to all samples to limit the variability of samples in multiple batches and mRNA was poly (A)-captured using NEXTflex Poly(A) beads (Bioo Scientific). The libraries were prepared using the NEXTflex Rapid Directional RNA-seq (RNA sequencing) Kit (Bioo Scientific) and were quantified and qualified using a Quantus quantification kit (Promega) and a fragment analyzer advanced analytical. Sequencing was performed on a NextSeq 500 Flow Cell High Output instrument (single read, 75 base pairs). Three independent data sets were obtained: PBMCs from day 2 to 10, PBMCs 1 DPI (PBMC1), and the organs. The first data set included 21 samples: three infection status (mock, Josiah, and AV) and three timepoints (2, 4, and 10 DPI for LASV-infected animals and 28 DPI for the mock infection). The second data set contained nine samples (three infection status, three animals per group). The last one consisted of 69 samples: three organs (liver, spleen, and LNs), three infection status (mock, AV, and Josiah), and three timepoints (2, 5, and 11 DPI for LASV-infected animals and 28 DPI for mock-, as well as for AV-infected NHPs).

Bioinformatics analysis was performed using the RNA-seq pipeline from Sequana[72]. Reads were cleaned of adapter sequences and low-quality sequences using cutadapt version 1.11[73]. Only sequences of at least 25 nucleotides in length were considered for further analysis. STAR version 2.5.0a[74], with default parameters, was used for alignment against the reference genome (*Macaca Fascicularis* 5 from ENSEMBL version 95). Reads were assigned to genes using featureCounts version 1.4.6-p3[75] from Subreads package (parameters: -t gene -g ID -O -s 2).

Statistical analyses were performed to identify genes for which the expression profiles were significantly different for each pair of biological conditions. For each data set (PBMC, PBMC1, and organs), genes exhibiting expression lower than one count per million for at least three samples were considered to have a low level of expression and discarded from the analysis. For the PBMC and PBMC1 data sets, differential analysis was performed using the DESeq2 R package (v1.24.0)[76]. The model was adjusted for the effect of infection status, timepoints, animal identifier, and sequencing lane, when relevant. For the organ data set, differential analysis was performed using the limma R package (v3.40.6)[77]. Expression values were normalized using voom transformation and the model was adjusted for the effect of

organ, infection status, timepoint, and animal identifier. For the organ data set, the sex of the animals was not included because of the statistical confounding effects of the experimental conditions of interest. For both data sets, gene set functional analysis was performed to identify gene sets and pathways enriched in the various biological conditions using the CAMERA function from the limma R package[78] based on the same model as in the differential analysis. The interrogated gene signatures included the Hallmark gene set and the Reactome and KEGG pathways from the Molecular Signature Data Base[79]. The probabilities of lethal infection and infection status were estimated using the RandomForest R package (v4.6-14)[80]. Each probability was computed for each figure by estimating a random forest for the gene expression subset for a specific number of genes (seven for lethal infection and six for infection status). The resulting probabilities were predicted by the model to assess the predictive ability of the selected biomarkers.

**Statistical analysis**. Statistical analyses of all collected data excluding transcriptomics were performed using GraphPad Prism 8 (GraphPad, La Jolla, CA, USA). Data of continuous variables are expressed as individual points or as the mean ± standard error of the mean. Student's $t$ test or one-way analysis of variance with multiple comparisons were used to compare the means between groups of data with a normal distribution. The Kruskal–Wallis multiple comparisons test was used to compare the means between groups of data with a non-normal distribution. The Shapiro–Wilk test was used to test for normality. The statistical analyses performed on the transcriptomic data sets are described in the appropriate Methods section. In addition, the sample sizes and number of replicates were described in the text or figure captions.

**Reporting summary**. Further information on research design is available in the Nature Research Reporting Summary linked to this article.

## Data availability
The data sets related to RNA sequencing generated during and/or analyzed during the current study are not publicly available owing to further investigation currently in progress but are available from the corresponding author on reasonable request.

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

## Acknowledgements

We thank Emmanuelle Guillot-Combe for invaluable support in the project. We are grateful to S. Mundweiler, B. Labrosse, D. Pannetier, S. Mély, A. Bocquin, D. Thomas, S. Godard, E. Moissonnier, A. Pocquet, J. Valois, and C. Léculier (P4 INSERM–Jean Mérieux, US003, INSERM) for assistance in conducting the BSL-4 experiments. We also thank M. Langry, C. Tournebize, and J. Noiret for their technical help during the in vivo experiments. We thank F. Chrétien, D. Hardy, P. Ave, and L. Fiette (Experimental Neuropathology Unit, Institut Pasteur) for training and advice on immunohistochemical processing and staining. We also thank O. Merabet, S. Krieger, C. Germain, and M. Jourdain (UBIVE) for technical help. We thank S. Becker (Institut of Virology, Marburg, Germany) for providing us with the LASV strains, T. G. Ksiazek, P. E. Rollin, and P. Jahrling (Special Pathogens Branch, Center for Disease Control, Atlanta, GA) for the LASV monoclonal antibodies, and H. Contamin (Cynbiose) for providing healthy primate blood. This study was funded by The Délégation Générale pour l'Armement (Agence Nationale de la Recherche - Accompagnement Spécifique des Travaux de Recherches et d'Innovation Défense, ANR-ASTRID 2014, France), the Fondation pour la Recherche Médicale (FRM, France), and a PhD grant attributed to N.B. from the Direction Générale de l'Armement (France, 2014015). The funders had no role in study design, data collection and analysis, the decision to publish, or preparation of the manuscript.

## Author contributions

S. Baize conceived the project. N.B., S.R., and S. Baize designed the protocols. N.B., S.R., A.J., M.M., X.C., J.S., and C.P. performed the RNA extractions, isolation of cells from organs, PBMC isolation, and flow cytometry sample preparation. L.B., S.B., A.V., A.D., F.J., and C.B. took care of the animals and performed the LASV challenge, scoring, sampling, hematological and biochemical analyses, and necropsies. L.B., F.J., and C.B. supervised the animal experiments. C.C. and H.R. supervised and validated the BSL-4 protocols. N.B., S. Baize, S.R., and A.J. performed the ELISA and Luminex assays. S.R. and S. Baize performed the flow cytometry analyses. N.B. and S.R. performed the RT-qPCR analyses on the plasma and organ samples. N.B. and S.R. performed the viral titrations and seroneutralization assays. N.B. prepared the samples for transcriptomic analyses. N.B. and J.H. performed the histological analyses. G.J. analyzed the anatomo-pathological changes. C.L.-L. and R.A. performed the RNA-seq. E.P., N.P., R.L., and M.-A.D. analyzed the transcriptomic data. N.B. and S. Baize analyzed the data and wrote the paper. S. Baize supervised the project.

## Competing interests

The authors declare that they have no competing interests.
