## [Peer Review File · Communications Biology]

Reviewers' comments:

Reviewer #1 (Remarks to the Author):

In this study, the authors leverage the translational cynomolgus macaque model to analyze the pathogenesis of mild and fatal Lassa virus (LV) infection at the biochemical, cellular, histological and transcriptional levels to identify potential predictors of disease outcome. Mild infection with the AV strain and fatal infection with the Josiah strain of LV both result in viral replication, induction of pro-inflammatory and anti-inflammatory cytokines, changes in the frequencies of lymphocytes, as well as transcriptional changes consistent with disease kinetics. However, fatal infection was associated with widespread viral dissemination from the site of infection, a cytokine storm, lymphopenia, neutrophilia and more severe transcriptional changes, some of which reflect those seen in septic shock syndrome. The authors conclude that fatal LV infection is associated with an imbalanced innate immune response and defective T cell response.

The strength of this study is the use of the cynomolgus macaque model to accurately recapitulate human disease pathology with sufficient numbers of animals per experimental and control groups. This study also employs a thorough array of techniques, including clinical data, biochemical data, flow cytometry, soluble mediator profiling and transcriptomics. Additionally, statistical analyses are rigorous. Limitations of this study include the over-interpretation of data presented, lack of data directly profiling innate immune cells, poor interpretation of transcriptional data, and failure to integrate the results of various approaches used to provide a cohesive picture of LV infection pathogenesis. Specific comments are provided below:

- Introduction and abstract
 - The relevance of the AV and Josiah strains to affected and at-risk populations should be clarified.
 - The strains used in this study and the basis of their differences (i.e. 82% nucleotide homology) should be specified.
 - The inoculation site of LV infection is not specified here or in the materials and methods. This is critical for interpreting data (i.e. draining lymph nodes can have distinct differences in responses compared to other tissues).
 - The novel insight that will be provided by this study (i.e. longitudinal transcriptional studies, soluble mediator profiling, etc. in a nonhuman primate model) needs to be clearly articulated.
 - The authors imply that their study develops a cynomolgus macaque model that recapitulates human disease, but this model was already established (references 9-11 in manuscript).
- Clinical parameters and viral spreading
 - Inoculation site should be specified.
 - For figure 1A, the authors should consider error bars for improved legibility of graphs.
 - Infectious virus and viral RNA was found in the bone marrow of infected animals and may explain findings from flow cytometry, such as lymphopenia, and transcriptional data.
 - The abbreviation "SLO" is introduced (line 111) without clarification.
- Histopathological features of LF
 - The authors should clarify tingible-body macrophages.
 - Key histological features, such as foci in the adrenal glands (figure 2g), should be emphasized with arrows and other drawn indicators.
- Biochemical analysis and soluble mediators during LF
 - Use of individual points to represent each animal (as in figure 1) should be considered for figure 3
 - Figures 3B-E are addressed in the text without casual references to their biological significance. For example, IL-6 and TNF-alpha are not articulated as pro-inflammatory, and perforin and granzyme B are not associated with cytotoxic activity. This is relevant for supporting the data and conclusions from figures 4-7.
- Adaptive immune response during Lassa fever
 - Kinetic data of total white blood cell count should be provided to provide a broader picture of leukocyte dynamics in each infection before analyzing specific cell subsets
 - Data for innate immune cells, including macrophages, dendritic cells and monocytes, are not presented despite the important role of these cells in LV infection. NK cell data are also not provided even though other data in this study suggest a role for cellular cytotoxicity in survival.
 - Activated plasma B cells (i.e. CD38+CD27+) should also be profiled to fully analyze the humoral response. There is skewed attention towards cellular T cell immunity.

- As stipulated above, how lymphopenia in Josiah infection may be related to viral dissemination to the bone marrow could be further investigated here with flow cytometry to provide a clearer description of LV pathogenesis and correlates of survival.
- The authors should take caution in analogizing the percent of T cells positive for perforin, granzyme, B, Ki67 or other markers of activation as T cell responses. This is not functional data, as compared to ELISPOT and intracellular cytokine staining following antigen stimulation.
- It is not clear if any statistical analyses were performed for figures 4C and 4 E.
- The choice of profiling only T cells in mesenteric lymph nodes and spleen is not validated. Although data suggest antibodies in the blood may not have a critical role in disease outcome (figure 3S), tissue data may suggest otherwise.
- Transcriptomic changes
 - The kits used to prepare libraries for sequencing and the sequencing platform are not specified here or in the methods.
 - There is no discussion of the biological relevance of differentially expressed (DE) genes with exception of several self-selected pathways. The kinetics of gene expression are the sole focus of this section. For gene enrichment terms, the p-values and/or FDR values for each term should be reported.
 - Transcriptomic data is not integrated with previous data presented by the author and is not represented in the context of LV infection pathogenesis.
 - Venn diagrams of DEs between each infection group (per time point or an aggregation of timepoints) should be shown to emphasize the differences and/or similarities (if any) in the composition of DEs in addition to the kinetics already presented in the manuscript.
 - Any discrepancies between flow cytometry data, cytokine profiling data and transcriptional data should be addressed. For instance, IL6 and TNFalpha are expressed at greater levels in Josiah infection (figure 5B), which agrees with figure 3, but the trend for IL-10 does not.
 - Figure 5A: the content of this heat map is not addressed. The meaning of these genes and ones appearing in subsequent panels are not explained. This heatmap should be made supplemental or should be discarded altogether if description of any key genes are not made.
 - Figure 5B-D and 6 A-C: The p-values and FDR values for these pathways (derived from Limma) need to be provided. The use of black and gray bars to depict different types of significance is not visually aesthetic although it is efficient. The authors should consider using color-coded asterisks beside genes. Data presented here would be best supported by existing and additional (i.e. monocytes) flow cytometry data.
 - Figure 6: Lymph nodes examined here are not specified.
 - Figure 6D: Fold change of expression between 10 and 0 DPI should be used for this figure. It is not clear whether these sepsis-related genes are also DE genes in this study or if there are significant changes in fold change.
 - Figure 7A: DEs identified here to predict infection vs mock states are not specific to LF infection. It is unclear how this data would be clinically relevant.
 - Figure 7B: Like in figure 7A, the biological relevance of the identified DE genes needs to be stated.
 - Figure 7C: The longitudinal expression of these genes found at 1 DPI should be provided and, as above, be described in context of their biological relevance.
- Discussion should be re-organized to focus on integrating datasets to provide biological relevance of the findings. For instance, soluble mediator profiling data would benefit from an integration with flow cytometry data. Similarly, transcriptomic data report only on kinetics of selected genes and fails to provide biological relevance for these data and how they support and/or contradict flow cytometry data, cytokine data, etc.

Reviewer #3 (Remarks to the Author):

In this manuscript, the authors present a detailed study comparing infections of macaques with

either Lassa Josiah, causing severe infection, or with Lassa AV causing mild infection, and mock infections. They present extensive data showing up- and down-regulation of infection and immunity, of transcriptome profile markers and of blood chemistries over a 28 day period. This reviewer is very impressed with the amount and quality of this work. It is a very valuable contribution to the literature. However, the density of the data in figures sometimes makes the manuscript difficult to read. Perhaps it would be easier to read if the pie-charts of T cell subsets and the Log2 graphs of individual marker fluctuations would be placed in the supplementary data section.

The most valuable aspect of this manuscript is the comparison of severe and mild infections to no infection at all. Such a comparison allows one to see that infection with LASV AV strain induces better activation of T cell responses than LASV Joshia, and that the virulent infection has a clearly dysregulated inflammatory response reflected in disorganized lymph nodes and a profile similar to bacterial sepsis. The mild infection has a self-limited inflammatory response. This type of profile was also described by Djavani et al, 2007 for macaques infected with a virulent arenavirus and a mild arenavirus almost 13 years earlier. In the Djavani study, myeloid marker CD14 was strongly up-regulated early in the severe infection.

The importance of these studies is that the gene expression differences between mild and severe infections were seen by day 2 (prior to the detection of viremia) and could be used to predict the outcome of infection. Several publications prior to the Baize study contributed to the conceptual portent of its message and were not mentioned.

The authors will find some specific comments as sticky notes in the PDF file attached.

Responses to the referees

We are grateful to the referees for their helpful and relevant comment. We have made our best to answer them.

Reviewer #1

- *Introduction and abstract*

- *The relevance of the AV and Josiah strains to affected and at-risk populations should be clarified.*

A few sentences have been added in the introduction (p 3)

- *The strains used in this study and the basis of their differences (i.e. 82% nucleotide homology) should be specified.*

Idem p 3

- *The inoculation site of LV infection is not specified here or in the materials and methods. This is critical for interpreting data (i.e. draining lymph nodes can have distinct differences in responses compared to other tissues).*

The inoculation site was the back of the thigh, and now indicated p 6 and 29

- *The novel insight that will be provided by this study (i.e. longitudinal transcriptional studies, soluble mediator profiling, etc. in a nonhuman primate model) needs to be clearly articulated.*

A sentence has been added in the introduction (p 5)

- *The authors imply that their study develops a cynomolgus macaque model that recapitulates human disease, but this model was already established (references 9-11 in manuscript).*

We have accordingly modified this point in the abstract (replacing *developed by used*)

- *Clinical parameters and viral spreading*

- *Inoculation site should be specified.*

See above

- *For figure 1A, the authors should consider error bars for improved legibility of graphs.*

Done, figure and legend modified accordingly

- *Infectious virus and viral RNA was found in the bone marrow of infected animals and may explain findings from flow cytometry, such as lymphopenia, and transcriptional data.*

A sentence has been added in the discussion (p 20) to link the tropism for BM with lymphopenia

- *The abbreviation "SLO" is introduced (line 111) without clarification.*

Done p 7

- *Histopathological features of LF*

- *The authors should clarify tingibile-body macrophages.*

This has been done p 8 by adding a sentence after the first citation of tangible-body macrophages in the text: '*it means macrophages containing many phagocytized, apoptotic cells in various states of degradation*'

- *Key histological features, such as foci in the adrenal glands (figure 2g), should be emphasized with arrows and other drawn indicators.*

Done, figure and legend modified accordingly

- *Biochemical analysis and soluble mediators during LF*

○ *Use of individual points to represent each animal (as in figure 1) should be considered for figure 3*

Done, figure and legend modified accordingly

○ *Figures 3B-E are addressed in the text without casual references to their biological significance. For example, IL-6 and TNF-alpha are not articulated as pro-inflammatory, and perforin and granzyme B are not associated with cytotoxic activity. This is relevant for supporting the data and conclusions from figures 4-7.*

Biological significance has been added for these data (p 10)

● *Adaptive immune response during Lassa fever*

○ *Kinetic data of total white blood cell count should be provided to provide a broader picture of leukocyte dynamics in each infection before analyzing specific cell subsets*

A graph has been added in figure 4a and was described in the result section (p 11)

○ *Data for innate immune cells, including macrophages, dendritic cells and monocytes, are not presented despite the important role of these cells in LV infection. NK cell data are also not provided even though other data in this study suggest a role for cellular cytotoxicity in survival.*

We agree with the reviewer that flow cytometry results on monocytes and dendritic cells would have been very interesting and useful. However, these data have only been partially acquired using fresh PBMC. Performing analysis with frozen PBMC is not possible as most of these cells are deleted during freezing process and biased results would be obtained. We have therefore added some data obtained with fresh cells in figure 4a, it means absolute numbers of mDC and monocytes as well as percentage of CD80+, CD86+, and CD40+ monocytes. We were not able to obtain reliable results for plasmacytoid DC nor for activation markers of mDC. We have analyzed the number of circulating NK cells (CD8+ CD3- CD20-), as well as their phenotype (KI67+, CD107a+, NKp80+, and NKG2D+) using frozen PBMC. These data are now presented in figure 4b.

○ *Activated plasma B cells (i.e. CD38+CD27+) should also be profiled to fully analyze the humoral response. There is skewed attention towards cellular T cell immunity.*

The phenotype of circulating B cells has been analyzed using frozen PBMC, splenocytes, and lymph node cells and added in figure 5c. In particular, we now presented the percentage of circulating naïve/unconventional memory B cells (CD20+ CD38mid CD27- CD10-), of transitory memory B cells (CD20+ CD38 mid CD10+ CD27-), of conventional memory B cells (CD20+ CD38mid CD27+ CD10-), and of plasma B cells (CD20low CD38bright CD27+) among B cells (CD20+) is presented, as well as the percentage of KI67+ cells among B cell subpopulations.

○ *As stipulated above, how lymphopenia in Josiah infection may be related to viral dissemination to the bone marrow could be further investigated here with flow cytometry to provide a clearer description of LV pathogenesis and correlates of survival.*

No analysis of bone marrow has been performed with flow cytometry. It is not possible to do it as supplementary experiment as we did not store bone marrow cells in nitrogen.

○ *The authors should take caution in analogizing the percent of T cells positive for perforin, granzyme, B, Ki67 or other markers of activation as T cell responses. This is not functional data, as compared to ELISPOT and intracellular cytokine staining following antigen stimulation.*

We fully agree with this comment. A sentence has been added in the discussion (p21) to indicate that quantification of cytokines produced by T cells after peptide stimulation would be necessary to analyze LASV-specific T cells.

○ *It is not clear if any statistical analyses were performed for figures 4C and 4 E.*

No statistic has been performed for these data. This has been described in the legend and these data are now supplementary.

○ *The choice of profiling only T cells in mesenteric lymph nodes and spleen is not validated.*

Although data suggest antibodies in the blood may not have a critical role in disease outcome (figure 3S), tissue data may suggest otherwise.

We have further analyzed immune cell population in SLO using frozen splenocytes and lymph node cells. As mentioned above, it was not possible to focus on macrophages and dendritic cells in frozen material, we therefore have only focused our flow cytometry study on NK and B cells. These data are now presented in figure 4c for NK cells and in figure 5c for B cells.

- *Transcriptomic changes*

○ *The kits used to prepare libraries for sequencing and the sequencing platform are not specified here or in the methods.*

A paragraph has been added in the according section of methods (p 34).

○ *There is no discussion of the biological relevance of differentially expressed (DE) genes with exception of several self-selected pathways. The kinetics of gene expression are the sole focus of this section. For gene enrichment terms, the p-values and/or FDR values for each term should be reported.*

We have added the p values in the respective figures and the FDR values in an excel file provided as supplementary material. We have made our best to further discuss the biological relevance of the DE genes and pathways.

○ *Transcriptomic data is not integrated with previous data presented by the author and is not represented in the context of LV infection pathogenesis.*

As above, we have improved the discussion about the links between the different data and LASV pathogenesis.

○ *Venn diagrams of DEs between each infection group (per time point or an aggregation of timepoints) should be shown to emphasize the differences and/or similarities (if any) in the composition of DEs in addition to the kinetics already presented in the manuscript.*

Venn diagrams are now presented in Fig S5 instead of the supplementary table 1.

○ *Any discrepancies between flow cytometry data, cytokine profiling data and transcriptional data should be addressed. For instance, IL6 and TNFalpha are expressed at greater levels in Josiah infection (figure 5B), which agrees with figure 3, but the trend for IL-10 does not.*

Concerning IL-10, the results can be explained by the fact that the source of IL-10 released in plasma was not PBMC but rather tissues/organs. Indeed, the transcriptomic data observed in liver are consistent with the plasma levels of IL-10 measured in Josiah-infected animals. A sentence has been added to discuss this point (p25).

○ *Figure 5A: the content of this heat map is not addressed. The meaning of these genes and ones appearing in subsequent panels are not explained. This heatmap should be made supplemental or should be discarded altogether if description of any key genes are not made.*

We fully agree with this comment. These heatmaps are now addressed in the result section (p15).

○ *Figure 5B-D and 6 A-C: The p-values and FDR values for these pathways (derived from Limma) need to be provided. The use of black and gray bars to depict different types of significance is not visually aesthetic although it is efficient. The authors should consider using color-coded asterisks beside genes. Data presented here would be best supported by existing and additional (i.e. monocytes) flow cytometry data.*

The *p* values are now presented for each gene set (fig 7b, 8abc). FDR values are presented in an excel file (supplementary file). We agree that the black and grey bars are not very aesthetic. However, we have not found a better representation for them. As pointed out by the reviewer, their use is nevertheless efficient and we therefore chose to keep them as color-coded asterisks would probably be more complex and confusing.

○ *Figure 6: Lymph nodes examined here are not specified.*

Done, it was the mesenteric lymph nodes, and the acronym LN has been replaced by MLN accordingly.

○ *Figure 6D: Fold change of expression between 10 and 0 DPI should be used for this figure. It is not clear whether these sepsis-related genes are also DE genes in this study or if there are significant changes in fold change.*

It is not possible to use the fold change between 10 and 0 DPI for infected animals, as no analysis of mRNA expression has been done at day 0. We only have analyzed post-infection samples and compared them with mock animals. We think that our method of comparison is acceptable as we analyzed three animals per group and performed a robust statistical analysis on data sets.

The sepsis-related genes listed in figure 7d and e are significantly DE in this study. The sentence has been modified accordingly p17.

○ *Figure 7A: DEs identified here to predict infection vs mock states are not specific to LF infection. It is unclear how this data would be clinical relevant.*

Indeed, the identified DE genes used to predict infection states are not specific for LASV, as already discussed p26 (*The analysis of the expression of these genes by RT-PCR could be used for diagnostic purposes, as previously proposed for other genes, assuming that they will likely not be specific for LF but rather represent an acute viral infection signature. Nevertheless, during an outbreak, such an approach would help to determine whether a subject has been infected very early after a high-risk contact and provide an idea of the severity of the coming disease. Whether the same sets of genes are modulated during human LF is yet to be determined, but similar early transcriptomic profiles are probably induced in the PBMCs of patients*). We have added a sentence explaining that the relative specificity for LF of these gene sets could also be challenged against other infectious diseases (p27). Therefore, these data would be clinically relevant during outbreaks in high-risk contacts or after laboratory exposure.

○ *Figure 7B: Like in figure 7A, the biological relevance of the identified DE genes needs to be stated.*

On the contrary to the determination of the infectious state, the prognostic markers are supposed to be used in patients with a confirmed LASV infection. Therefore, the lack of

specificity for LASV of the defined markers will not be an issue. We have added a reference with similar approach for dengue virus infection and a sentence p24: "The prognostic of LF severity could be useful in epidemic setting when intensive care can be provided to a limited number of patients".

○ *Figure 7C: The longitudinal expression of these genes found at 1 DPI should be provided and, as above, be described in context of their biological relevance.*

Transcriptomic changes measured at 1 DPI could not be compared with other samples as the sequencing of these mRNA was not performed together with the others. This has been explained in the figure legend.

● *Discussion should be re-organized to focus on integrating datasets to provide biological relevance of the findings. For instance, soluble mediator profiling data would benefit from an integration with flow cytometry data. Similarly, transcriptomic data report only on kinetics of selected genes and fails to provide biological relevance for these data and how they support and/or contradict flow cytometry data, cytokine data, etc.*

The discussion has been modified accordingly, with a particular emphasis in integrating the data obtained for soluble mediators with flow cytometry and transcriptomic results. The discrepancies and in contrast, the similarity between the different approaches have been highlighted.

Reviewer #2

The modifications contained in the PDF document provided by the reviewer have been made. In particular, we have added the proposed references.

The figures have been modified according to the reviewer's requests.

The pie-chart describing the memory T-cell responses are now presented as supplementary material (Fig S4).

Finally, we have added several references that described early differences in gene expression and their possible use as diagnostic and prognostic tools.

REVIEWERS' COMMENTS:

Reviewer #1 (Remarks to the Author):

The authors have adequately addressed concerns raised during the previous submission. The new myeloid cell data is a great addition to the manuscript.